# Directed-Tokens: A Robust Multi-Modality Alignment Approach to Large Language-Vision Models

**Thanh-Dat Truong[1], Huu-Thien Tran[1], Thai-Son Tran[2], Bhiksha Raj[3], Khoa Luu[1]**

[1]CVIU Lab, University of Arkansas, USA
[2]Vietnam National University, Ho Chi Minh City University of Science, Vietnam
[3]Carnegie Mellon University, USA

{tt032, ht035, khoaluu}@uark.edu, ttson@fit.hcmus.edu.vn, bhiksha@cs.cmu.edu
https://uark-cviu.github.io/projects/DirectedTokens

## Abstract

Large multimodal models (LMMs) have gained impressive performance due to their outstanding capability in various understanding tasks. However, these models still suffer from some fundamental limitations related to robustness and generalization due to the alignment and correlation between visual and textual features. In this paper, we introduce a simple but efficient learning mechanism for improving the robust alignment between visual and textual modalities by solving shuffling problems. In particular, the proposed approach can improve reasoning capability, visual understanding, and cross-modality alignment by introducing two new tasks: reconstructing the image order and the text order into the LMM's pre-training and fine-tuning phases. In addition, we propose a new directed-token approach to capture visual and textual knowledge, enabling the capability to reconstruct the correct order of visual inputs. Then, we introduce a new Image-to-Response Guided loss to further improve the visual understanding of the LMM in its responses. The proposed approach consistently achieves state-of-the-art (SoTA) performance compared with prior LMMs on academic task-oriented and instruction-following LMM benchmarks.

## 1 Introduction

Large multimodal models (LMMs) have gained more attention recently due to their outstanding capabilities in general-purpose assistants. By training via visual instruction tuning [39, 37, 38, 3, 29, 28], these LMMs, e.g., LLaVA [39], InstructBLIP [29, 28], MiniGPT-4 [69], Qwen-VL [4], have shown impressive performance on instruction-following and visual reasoning tasks. Then, these LMMs have been further developed for specific scientific tasks, e.g., LLaVA-Med [26], DriveGPT4 [56], PMC-LLaMA [53], etc. To measure the performance and capabilities of LMMs, several benchmarks [63, 58] have been introduced to evaluate the LMMs in different aspects, e.g., art, business, health and medicine, science, tech and engineering, and other fields. Recent studies further improve the performance of LMMs by scaling the training data [37, 38, 66, 65], using better visual encoders [5, 28, 28, 51] or large language models [25], improving objective learning [50, 31, 60, 68], using multimodal preference data [54, 45, 61], extending to other modalities [35, 34, 33] (e.g., videos [35, 34, 43], graphs [33]).

**Motivation of this Work.** While recent efforts in improving the performance of LMMs focus on scaling data and models [37, 38, 25, 65, 5, 28, 28, 51, 42, 49], *this work studies the pitfalls of these LMMs in a new simple but efficient aspect* (Fig. 1). In particular, LMMs are usually biased to language preferences since the large language model (LLM) has been pre-trained on the exascale

39th Conference on Neural Information Processing Systems (NeurIPS 2025).

data. Meanwhile, due to the data complexity, the LMMs tend to overlook the information of visual inputs. For example, an LMM can achieve similar results using only languages [50].

**How does visual information impact LMM's responses?** To demonstrate this point, we conduct experiments by evaluating the LMM, i.e., LLaVA v1.5 7B [37], on the ScienceQA-IMG [41] and MMMU [63] benchmarks. We remove the visual information by using the black image as an input instead of the original image of the benchmark. As in Table 1, the performance of the models is still maintained without the visual information. As in Fig. 2, LLaVA [37] answers similarly for two cases even though the important information of the images in the second case was blacked out. The model cannot learn a well-alignment between visual and textual features. Thus, the answers produced by the LLaVA model are dominated by the language model with low consideration of visual inputs. This problem indicates the multimodal alignment in current LMMs has not been well learned yet, leading to prioritizing language preferences and overlooking the visual information. As a result, the LMMs will produce less informative outputs or even hallucinate the results, leading to low model performance.

In this paper, we, therefore, address two fundamental questions for the current LMMs. For the first question, given an image whose patches are shuffled, *will the LMM be able to reconstruct the original image matched with language representations?* If this is not the case, the LMM relies on the language encoder, which ignores the visual information. Then, in the second question, given a shuffled textual description of images, *will the LMM be able to reconstruct the textual sentence to represent the visual information in the image?*

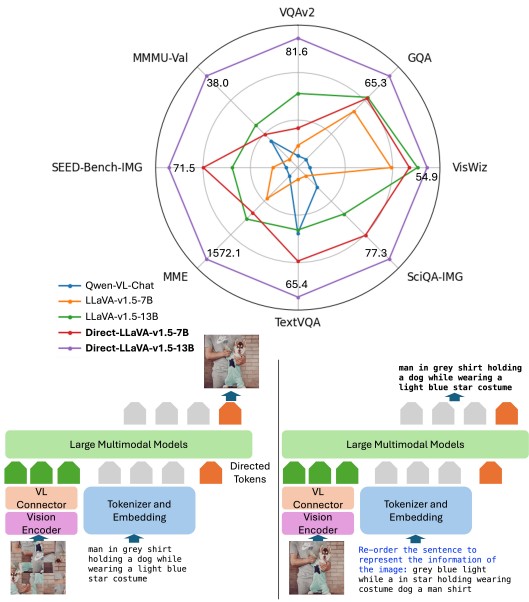

Figure 1: Our Proposed **Direct-LLaVA** achieves State-of-the-Art performance on various LMM benchmarks (Top) with new shuffle learning approaches of Image (Bottom Left) and Text (Bottom Right) during pre-training and fine-tuning phases.

If the LMM cannot do that, it means the alignment between visual and textual features has not been well represented by the LMM. Finally, by addressing these two questions, we introduced a novel learning approach to improve the robustness of the LMM models.

**Contributions of this Work.** This paper presents a novel learning approach to improving the alignment between visual and textual features in the LMM via solving the shuffle problems. Our approach adopts a simple but efficient learning strategy to learn the right order of visual information and textual descriptions, thus improving their alignment. Our contributions

Figure 2: The Example of Answer Produced by LLaVA v1.5 [37]. The first two rows are samples selected from ScienceQA-IMG, and the last two rows are sampled from MMMU.

can be summarized as follows. First, we introduce the problem of ordering the visual and textual information in LMM. In particular, we introduce a new shuffle learning method in the pre-training and fine-tuning phase, forcing the model to reconstruct the right order of images and textual descriptions for better alignment. Second, to support shuffle learning, we propose a new directed-token approach to capture visual and textual correlations in the LMM, thus improving the capability of reconstructing the right order of visual inputs. Third, to improve the mutual information between visual and textual features, we introduce a new Image-to-Response Guided loss based on the attention layers to enhance the information flow of the visual inputs to the LMM's responses. Finally, our ablation studies have shown the effectiveness of different aspects of our approach. Through inten-

sive experiments, our approach has achieved state-of-the-art performance on academic task-oriented and instruction-following LMM benchmarks.

## 2 Related Work

**Large Multimodal Models.** Thanks to the remarkable advancements in LLMs [34, 10, 1, 46, 3], it has promoted the development of LMMs. Numerous LMMs have been developed to enhance effectiveness in handling such data, which can be classified into large vision-

Table 1: Performance of LLaVA v1.5 [37] With and Without Visual Information.

| LMM | Image | SciQA-IMG | MMMU-Val |
|---|---|---|---|
| LLaVA-v1.5-7B | Origin | 66.8 | 35.3 |
| LLaVA-v1.5-7B | Black | 64.1 | 32.4 |

language models [39, 37], large video-language models [52, 67, 35, 30], and large audio-language models [20, 15]. Early work by [2] effectively bridged vision and language modalities in a few-shot learning setting, followed by [28], which enhanced inter-modality connectivity via Q-Former. This was further developed into an instruction-aware model within the vision-language instruction-tuning framework [39]. LLaVA [39] established a streamlined visual-to-language space projection using a linear layer, later refined by [37] with an MLP and AnyRes, a technique adept at handling high-resolution images. Subsequent studies [38, 24, 64, 23, 27] contributed further improvements, culminating in a robust model [25] capable of handling diverse vision tasks, including video comprehension. Later, [26] extended LLaVA to a biomedical model through a cost-effective curriculum learning approach. [7] leveraged multimodal federated learning to enhance LMM training. Meanwhile, [57, 55, 36] introduced LMMs for 3D point cloud understanding tasks.

**Shuffle Learning.** Decomposing an image into smaller patches, shuffling these patches, and then reconstructing their original order forms a classical pattern recognition problem known as the jigsaw puzzle. The early work in classification [14] solved the jigsaw puzzle by predicting the spatial position of each patch. Later, it has proven highly useful in self-supervised learning for feature representation. Indeed, [6] incorporates a jigsaw puzzle challenge alongside classification to improve visual information generalization across domains. [12] employs a jigsaw puzzle generator to create varying levels of granularity for fine-grained classification. Chen et al. [9] extended shuffle learning to transformers to enhance performance in visual recognition tasks. Truong et al. [48] developed a robust video understanding model by solving the shuffled temporal frames via the directed attention mechanism. This body of work suggests that using the jigsaw puzzle as a pretext task is broadly effective for generalizing spatial information within images. Building on this, we investigate the impact of reconstructing the order of elements not only in images but also in the texts, analyzing the interplay between these two modalities in relation to each other's permutated version.

## 3 The Proposed Directed Token Approach to LMM

Inspired by LLaVA [39, 37], we develop a new LMM, namely **Direct-LLaVA** (Fig. 3), by adopting the design of LLaVA v1.5. In particular, our Direct-LLaVA model consists of the vision encoder and the large language model. The visual features will undergo the vision-language (VL) connector to align with the textual features. Formally, given the image $\mathbf{x}$ and a multi-turn conversation data $\left(\mathbf{x}_q^1, \mathbf{x}_a^1, \mathbf{x}_q^2, \mathbf{x}_a^2, ..., \mathbf{x}_q^M, \mathbf{x}_a^T\right)$ where $T$ is the number of turns, following the standard protocol of

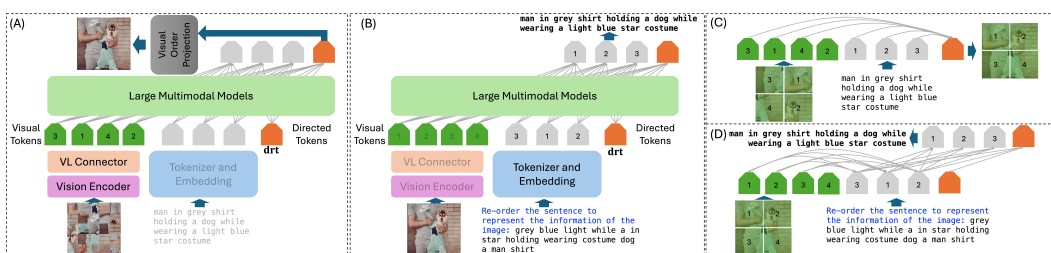

Figure 3: The Proposed Direct-LLaVA Framework. (A) Reconstructing Image Order Task. (B) Reconstructing Text Order Task. (C) Directed-Token Modeling Approach in Image Order Reconstruction. (D) Autoregressive Modeling Approach in Text Order Reconstruction.

[39, 37], the instruction data $\mathbf{x}_{\text{instruct}}^t$ in the $t^{th}$ turn is formed as in Eqn. (1).

$$\mathbf{x}_{\text{instruct}}^t = \begin{cases} \text{Randomly Choose } [\mathbf{x}_q^1, \mathbf{x}] \text{ or } [\mathbf{x}, \mathbf{x}_q^1] & \text{If } t = 1 \\ \mathbf{x}_q^t & \text{If } t > 1 \end{cases} \tag{1}$$

Then, learning LMM can be formed as an auto-regressive training objective as in Eqn. (2).

$$\theta^* = \arg\max_\theta \mathbb{E}_{\mathbf{x}_a, \mathbf{x}, \mathbf{x}_{\text{instruct}}} \log p(\mathbf{x}_a | \mathbf{x}, \mathbf{x}_{\text{instruct}})$$

$$= \arg\max_\theta \mathbb{E}_{\mathbf{X}_a, I, \mathbf{x}_{\text{instruct}}} \sum_{i=1}^{L} \log p_\theta(x_i | \mathbf{x}, \mathbf{x}_{\text{instruct}<i}, \mathbf{x}_{a<i}) \tag{2}$$

where $L$ is the sequence length of the answer $\mathbf{x}_a = [x_1, x_2, ..., x_L]$, $\theta$ is the parameters of the LMM model, and $\mathbf{x}_{\text{instruct}<i}$ and $\mathbf{x}_{a<i}$ are the tokens of instructions and answers in all turns before token $x_i$. Similar to LLaVA [39, 37], Direct-LLaVA follows two stages of the instruction-tuning procedure, i.e., pre-training and fine-tuning.

The success of the current state-of-the-art LMMs [39, 37, 38] relies on auto-regressive modeling. Indeed, the form of auto-regression naturally matches the nature of the data, where each input token in the sequence depends on the previous ones. This modeling approach can capture the complex dependencies and correlations within the image and language and maintain consistency and coherence in data. As a result, the order of data, i.e., visual image patches or textual sentences, plays an important role since the LMM auto-regressively models multimodal inputs conditioning on all previous elements to maintain context and coherence. If the data order is incorrect, the LMM loses its ability to model correct contexts and produce logically consistent predictions. Under this modeling principle, we aim to improve the LMMs by addressing two fundamental questions. First, given a random shuffled image, we aim to develop an LMM capable of reconstructing the original image that is represented by the current textual descriptions (Fig. 3(A)). This learning mechanism encourages the LMM to learn the strong correlation between visual and textual features, thus providing a better understanding of visual information. Second, given an original image and a shuffled textual description, our LMM aims to predict the correct textual description that represents the visual information of the image (Fig. 3(B)). The learning objective allows the model to improve correlation across modalities further, thus enhancing the important role of visual information in the LMM's responses. In the next sections, we will describe our approach to developing these two learning objectives in the pre-training and fine-tuning phases of the LMMs.

### 3.1 The Shuffle Learning In Pre-training Phase

The primary goal of the pre-training phase is to learn the alignment between the visual and the language spaces. The traditional LMM [39, 37] train the alignment based on the single-turn conversations (Fig. 4(A)) where the instruction data is formed as in Eqn. (3).

$$\begin{cases} \mathbf{x}_{\text{instruct}} & = \text{Randomly Choose } [\mathbf{x}_q, \mathbf{x}] \text{ or } [\mathbf{x}, \mathbf{x}_q] \\ \mathbf{x}_a & = \mathbf{p} \end{cases} \tag{3}$$

where $\mathbf{x}_q$ is randomly sampled from a set of questions designed to ask for the content of an image, and $\mathbf{p}$ is the textual description of the image $\mathbf{x}$. Then, learning the LMM with single-turn conversation can be formed as in Eqn. (4).

$$\theta^* = \arg\max_\theta \mathbb{E}_{\mathbf{x}_a, \mathbf{x}, \mathbf{x}_{\text{instruct}}} \log p(\mathbf{x}_a | \mathbf{x}, \mathbf{x}_{\text{instruct}}) \tag{4}$$

**Reconstructing Image Order Task.** To improve the visual understanding of the LMM, we introduce a new image order reconstructing task during the pre-training phase. In particular, during training, for each image $\mathbf{x}$, we will randomly shuffle image patches, followed by reconstructing the original order of the image described via the textual description $\mathbf{p}$ via the LMM. Formally, let $\bar{\mathbf{x}} = \mathcal{P}(\mathbf{x}, \mathbf{k})$ be the shuffled image where $\mathcal{P}$ is the permutation method that shuffles the patches of the image (we use the patch size of the vision encoder) and $\mathbf{k}$ is the indexing associated with the permutation. In this learning task, the textual description of the image is crucial for reconstructing the original image from the shuffled version. It provides additional reference knowledge that supports the LMM in learning to correct the image order. As shown in (Fig. 4(B)), while both predictions

are meaningful images, only one of them is correct and matches the description. Therefore, the instruction data in this image ordering task can be formed as follows,

$$\mathbf{x}_{\text{instruct}}^{\text{image}} = \text{Randomly Choose } [\mathbf{p}, \bar{\mathbf{x}}] \text{ or } [\bar{\mathbf{x}}, \mathbf{p}] \tag{5}$$

Then, learning to reconstruct the order of the image via the LMM model can be formulated as in Eqn. (6).

$$\theta^* = \arg\max_{\theta} \mathbb{E}_{\mathbf{k}, \bar{\mathbf{x}}, \mathbf{x}_{\text{instruct}}^{\text{image}}} \log p(\mathbf{k} | \bar{\mathbf{x}}, \mathbf{x}_{\text{instruct}}^{\text{image}}) \tag{6}$$

**Directed Tokens.** To predict the permutation index $\mathbf{k}$ from the shuffled image, we propose to design a visual order projection head via a linear layer. We can adopt the last hidden-stage features of the visual tokens as the input of this projection head. However, this approach is inefficient for two reasons. First, the visual tokens are designed to contain the general knowledge of the visual inputs. Second, if the visual tokens are placed at the beginning of the instruction input (i.e., $[\bar{\mathbf{x}}, \mathbf{p}]$), the textual information from the description is not captured by the visual tokens via attention mechanisms due to the autoregressive modeling of the LMM (in this case, the texts are future tokens of the visual tokens). Therefore, to address this problem, we introduce **a new learnable directed token drt** placed at the end of the input token sequence (Fig. 3(C)). The term "directed tokens" pays tribute to concepts of the order of the input tokens in shuffle learning tasks.

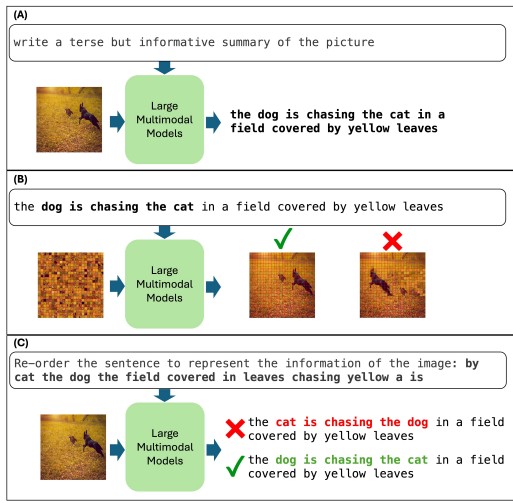

Figure 4: (A) Traditional Single-Turn Conversation Task. (B) Reconstructing Image Ordering Task. (C) Reconstructing Text Ordering Task.

Then, the permutation index $\mathbf{k}$ can be predicted by using the hidden-stage features of $\mathbf{drt}$ as $\hat{\mathbf{k}} = \text{ProjectLN}(\mathbf{drt}_h)$, where $\mathbf{drt}_h$ is the last hidden-stage feature of the directed token, and ProjectLN is the order projection layer. The instruction data with the directed token is formed as,

$$\mathbf{x}_{\text{instruct}}^{\text{image}} = \text{Randomly Choose } [\mathbf{p}, \bar{\mathbf{x}}, \mathbf{drt}] \text{ or } [\bar{\mathbf{x}}, \mathbf{p}, \mathbf{drt}] \tag{7}$$

Since the directed token **drt** *is placed at the end of instruction data*, it can efficiently capture the knowledge from both visual and textual features via the self-attention mechanism with autoregressive modeling. Therefore, the directed token can provide better information from visual and textual knowledge to predict the permutation index than the visual tokens of the image.

**Reconstructing Text Order Task.** Learning LMM requires a deep understanding of the correlation between visual and textual features. To further improve the visual understanding and reduce the dominance of the language modality of the LMM, we present a new reconstructing text order task during the pre-training phase. Formally, given an original image $\mathbf{x}$ and a shuffled word-order description $\bar{\mathbf{p}}$, the goal of this task is to encourage the LMM to predict the back of the original description matched with the visual information represented in the image. Although this task can be easily accomplished using the large language model of the LMM without visual features, visual information plays an important role in improving the visual understanding of the LMM in this learning paradigm. For example, as shown in Fig. 4(C), while both reconstructed sentences predicted by the LMM are meaningful, only one is correct with the context represented by the visual information. In this context, the image plays a significant role since it provides an additional reference that aids the LMM to accurately re-order the shuffled descriptions. Therefore, the LMM must incorporate and understand the visual features to correctly reconstruct the text order, as shown in Fig. 3(D). In this task, the instruction data can be formulated as in Eqn. (8).

$$\begin{cases} \mathbf{x}_{\text{instruct}}^{\text{text}} &= \text{Randomly Choose } [\mathbf{q}, \bar{\mathbf{p}}, \mathbf{x}] \text{ or } [\mathbf{x}, \mathbf{p}, \bar{\mathbf{p}}] \\ \mathbf{x}_a^{\text{text}} &= [\mathbf{p}, \mathbf{drt}] \end{cases} \tag{8}$$

where $\mathbf{q}$ is the prompt that instructs the reconstructing text order task, i.e., $\mathbf{q} = \texttt{re-order the}$ $\texttt{sentence to represent the information of the image:}$. It should be noted that although

the directed token $\mathbf{drt}$ may not contribute directly to text order predictions, $\mathbf{drt}$ is placed at the end of the answers to ensure consistency in the prompting format with the prior reconstructing image order task. Then, the text reconstructing order task via the LMM model can be formulated as follows,

$$\theta^* = \arg\max_\theta \mathbb{E}_{\mathbf{x}_a^{\text{text}},\mathbf{x},\mathbf{x}_{\text{instruct}}^{\text{text}}} \log p(\mathbf{x}_a^{\text{text}}|\mathbf{x}, \mathbf{x}_{\text{instruct}}^{\text{text}}) \tag{9}$$

**Learning Objective of the Pre-training Phase.** To summarize, our LMM model is optimized with three learning objectives, including the traditional single-turn conversation, the reconstructing image order task, and the reconstructing text order task. Formally, the learning objective of the pre-training phase can be formed as in Eqn. (10).

$$\theta^* = \arg\max_\theta \left[ \mathbb{E}_{\mathbf{x}_a,\mathbf{x},\mathbf{x}_{\text{instruct}}} \log p(\mathbf{x}_a|\mathbf{x}, \mathbf{x}_{\text{instruct}}) + \mathbb{E}_{\mathbf{k},\bar{\mathbf{x}},\mathbf{x}_{\text{instruct}}^{\text{image}}} \log p(\mathbf{k}|\mathbf{x}, \mathbf{x}_{\text{instruct}}^{\text{image}}) + \mathbb{E}_{\mathbf{x}_a^{\text{text}},\mathbf{x},\mathbf{x}_{\text{instruct}}^{\text{text}}} \log p(\mathbf{x}_a^{\text{text}}|\mathbf{x}, \mathbf{x}_{\text{instruct}}^{\text{text}}) \right] \tag{10}$$

## 3.2 The Shuffle Learning In Fine-tuning Phase

The goal of the fine-tuning phase is to enable visual and language understanding and a general-purpose visual assistant of the LMM through instruction-tuning with multi-round conversation. The training process of the LMM can be defined as in Eqn. (2) with the instruction data formed in Eqn. (1). This fine-tuning procedure helps the model improve the alignment between visual and textual modalities and enables language-driven visual reasoning. Then, to further improve the visual understanding and reasoning skills of the LMM, we adopt the **Reconstructing Image Order Task** presented in the previous section into the fine-tuning phase. This objective helps to improve the reasoning skills and understanding of the LMM by learning to reconstruct the image based on the content of the multi-round conversation instruction data. In this learning task, the directed token $\mathbf{drt}$ is placed at the end of the last answer to comprehensively capture the context of the entire conversation for the reconstructing image order task. Formally, the last answer of instruction data can be structured as $\mathbf{x}_a^T = [\mathbf{x}_a^T, \mathbf{drt}]$. Then, the learning objective of the fine-tuning phase with the reconstructing image order task can be formed as,

$$\arg\max_\theta \mathbb{E}_{\mathbf{x}_a,\mathbf{x},\mathbf{x}_{\text{instruct}}} \left[ \log p(\mathbf{x}_a|\mathbf{x}, \mathbf{x}_{\text{instruct}}) + \log p(\mathbf{k}|\bar{\mathbf{x}}, \mathbf{x}_{\text{instruct}}) \right] \tag{11}$$

where $\bar{\mathbf{x}} = \mathcal{P}(\mathbf{x}, \mathbf{k})$ is the shuffled version of the original image $\mathbf{x}$, and $\mathbf{k}$ is the permutation index. While the first objective enables the reasoning capability of the LMM, the second objective will improve the visual understanding of the LMM by reconstructing the correct order of the image.

It should be noted that we do not perform the reconstructing text order task during the fine-tuning phase, since the purpose of the fine-tuning task is to enable reasoning based on the multi-round conversation. The shuffled texts could destroy the context of the conversation and the reasonability of the LMM, leading to suboptimal performance. Therefore, we only adopt the reconstructing image order task during the fine-tuning phase to ensure the reasoning and visual understanding capability of the LMM.

## 3.3 Image-to-Response Guided Learning

Enhancing the visual knowledge in the response produced by the LMM is important since it will help the model capture more visual features, therefore enhancing the robustness of the answers. To further improve the visual understanding of the model and reduce the burden of the LMM when learning to capture visual and textual correlations, we propose a new Image-to-Response Guided loss to enforce the attention learning from the prior knowledge of image and response. Formally, the proposed **Image-to-Response Guided Loss** can be formulated as in Eqn. (12).

$$\mathcal{L}_{\text{I}\to\text{R}} = \frac{1}{L|\mathcal{V}||\mathcal{R}|} \sum_{l=1}^{L} \sum_{v\in\mathcal{V}} \sum_{r\in\mathcal{R}} (1 - \alpha_{v,r}^l) \tag{12}$$

where $v$ is the position of the visual token, where $r$ is the position of the response token, $\mathcal{V}$ is the list of visual tokens, $\mathcal{R}$ is the list of textual response tokens, $\alpha_{v,r}^l$ is the attention score from the visual token $v$ to the response token $r$ at the $l^{th}$ layer of the LMM, and $L$ is the number of layers in the LMM. Minimizing the Image-to-Response Guided loss will increase the attention score from the visual tokens to the response tokens produced by the LMM, i.e., $\alpha_{v,r}^l$. Therefore, it will help to indicate the attention learning to enhance the impact of the visual features in its textual responses.

Table 2: Comparison with prior methods on academic-task-oriented benchmarks.

| Method | LLM | Image Size | Data Size Pretrain | Data Size Finetune | VQAv2 | GQA | VizWiz | SciQA-IMG | TextVQA |
|---|---|---|---|---|---|---|---|---|---|
| BLIP-2 [28] | Vicuna-13B | $224 \times 224$ | 129M | - | 65.0 | 41.0 | 19.6 | 61.0 | 42.5 |
| InstructBLIP [11] | Vicuna-7B | $224 \times 224$ | 129M | 1.2M | - | 49.2 | 34.5 | 60.5 | 50.1 |
| InstructBLIP [11] | Vicuna-13B | $224 \times 224$ | 129M | 1.2M | - | 49.5 | 33.4 | 63.1 | 50.7 |
| Shikra [8] | Vicuna-13B | $224 \times 224$ | 600K | 5.5M | 77.4 | - | - | - | - |
| IDEFICS-9B [19] | LLaMA-7B | $224 \times 224$ | 353M | 1M | 50.9 | 38.4 | 35.5 | - | 25.9 |
| IDEFICS-80B [19] | LLaMA-65B | $224 \times 224$ | 353M | 1M | 60.0 | 45.2 | 36.0 | - | 30.9 |
| Qwen-VL [4] | Qwen-7B | $448 \times 448$ | 1.4B | 50M | 78.8 | 59.3 | 35.2 | 67.1 | 63.8 |
| Qwen-VL-Chat [4] | Qwen-7B | $448 \times 448$ | 1.4B | 50M | 78.2 | 57.5 | 38.9 | 68.2 | 61.5 |
| LLaVA-1.5-7B [37] | Vicuna-7B | $336 \times 336$ | 558K | 665K | 78.5 | 62.0 | 50.0 | 66.8 | 58.2 |
| LLaVA-1.5-13B [37] | Vicuna-13B | $336 \times 336$ | 558K | 665K | 80.0 | 63.3 | 53.6 | 71.6 | 61.3 |
| Direct-LLaVA | Vicuna-7B | $336 \times 336$ | 558K | 665K | 79.0 | 63.2 | 52.5 | 74.3 | 63.2 |
| Direct-LLaVA | Qwen-7B | $336 \times 336$ | 558K | 665K | 79.9 | 63.3 | 54.3 | 75.7 | 65.1 |
| Direct-LLaVA | LLaMA-8B | $336 \times 336$ | 558K | 665K | 80.0 | 63.9 | 54.4 | 75.8 | 64.9 |
| Direct-LLaVA | Vicuna-13B | $336 \times 336$ | 558K | 665K | **81.6** | **65.3** | **54.9** | **77.3** | **65.4** |

Finally, the Image-to-Response Guided loss will be integrated into the learning objective of the pre-training (Eqn. (10)) and fine-tuning phase (Eqn. (11)). In particular, the learning loss of the pre-training task $\mathcal{L}_{\text{pretrain}}$ can be formed as,

$$\mathcal{L}_{\text{pretrain}} = \mathcal{L}_{\text{CE}} + \mathcal{L}_{\text{Image-Order}} + \mathcal{L}_{\text{Text-Order}} + \mathcal{L}_{\text{I}\rightarrow\text{R}} \qquad (13)$$

Similarly, the learning objective of the fine-training task $\mathcal{L}_{\text{finetune}}$ can be formulated as in Eqn. (14).

$$\mathcal{L}_{\text{finetune}} = \mathcal{L}_{\text{CE}} + \mathcal{L}_{\text{Image-Order}} + \mathcal{L}_{\text{I}\rightarrow\text{R}} \qquad (14)$$

where $\mathcal{L}_{\text{CE}}$ is the typical loss of the LMM model [37, 39], $\mathcal{L}_{\text{Image-Order}}$ and $\mathcal{L}_{\text{Text-Order}}$ are the cross-entropy losses of the reconstructing image and text order tasks.

# 4 Experiments

## 4.1 Implementation and Benchmarks

**Implementation.** Our framework adopts the implementation of LLaVA v1.5 [37]. We use the CLIP-ViT-L-14 ($336^2$) encoder for the vision tower, and four different LLMs, i.e., Vicuna 7B [10], Vicuna 13B [10], Qwen 7B [3], and LLaMA3 8B [13]. We adopt the multi-layer perception [37] for the VL connector. To ensure the consistency of our implementation, the directed token **drt** is placed at the end of the sequence. We use 32 NVIDIA A100 in our experiments. For fair comparisons, we adopt the learning hyper-parameters of LLaVA v1.5 in our training. We use the training data of LLaVA v1.5 in our experiments. We also include additional ablation studies to analyze the impact of the different data size and other benchmarks in the supplementary.

**Shuffling Strategy.** Our image permutation function $\mathcal{P}$ shuffled the patches of the image where the patch size is $14 \times 14$. Therefore, it will be $N_P!$ permutations of the image patches where $N_P = \frac{336^2}{14^2} = 576$ is the number of image patches. Learning with all permutations remains inefficient due the its large variation. The choice of permutations plays an important role in improving visual understanding. In particular, if the two permutations are very far apart, the LMM may easily predict the image order since they have significant differences. Meanwhile, when all the permutations are close to each other, learning to reconstruct the order becomes a challenging problem since the two different permutations have minor differences. Therefore, to effectively develop a set of permutations, we randomly select $10,000$ permutations from $N_P = 576!$ so that the Hamming distance between two permutations is as close as possible. Meanwhile, the large language model can model complex semantic sentences. Thus, for the text shuffling, we randomly permute the word positions in the text sentences.

**Benchmarks.** Following the standard protocol [37], we evaluate our models on two sets of benchmarks, i.e., Academic Task-oriented and Instruction-Following LMM Benchmarks. The academic task-oriented task includes five benchmarks: Visual Question Answering V2 (VQAv2) [16], Question Answering on Image Scene Graphs (GQA) [18], Answer Visual Questions from People Who Are Blind (VizWiz) [17], Science Question Answering (SciQA-IMG) [41], and Visual Reasoning based on Text in Images (TextVQA) [47]. The Instruction-Following LMM has five benchmarks: Polling-based Object Probing Evaluation for Object Hallucination (POPE) [32], Multimodal

Table 3: Comparison with prior methods on benchmarks for instruction-following LMMs.

| Method | LLM | Image Size | Data Size | | POPE | | | SEED-Bench | | | MME | LLAVA-Wild | MM-Vet | MMMU-Val |
|---|---|---|---|---|---|---|---|---|---|---|---|---|---|---|
| | | | Pretrain | Finetune | rand | pop | adv | all | img | vid | | | | |
| BLIP-2 [28] | Vicuna-13B | 224 × 224 | 129M | - | 89.6 | 85.5 | 80.9 | 46.4 | 49.7 | 36.7 | 1293.8 | 38.1 | 22.4 | - |
| InstructBLIP [11] | Vicuna-7B | 224 × 224 | 129M | 1.2M | - | - | - | 53.4 | 58.8 | 38.1 | - | 60.9 | 26.2 | - |
| InstructBLIP [11] | Vicuna-13B | 224 × 224 | 129M | 1.2M | 87.7 | 77 | 72 | - | - | - | 1212.8 | 58.2 | 25.6 | - |
| IDEFICS-9B [19] | LLaMA-7B | 224 × 224 | 353M | 1M | - | - | - | - | 44.5 | - | - | - | - | - |
| IDEFICS-80B [19] | LLaMA-65B | 224 × 224 | 353M | 1M | - | - | - | - | 53.2 | - | - | - | - | - |
| Qwen-VL [4] | Qwen-7B | 448 × 448 | 1.4B | 50M | - | - | - | 56.3 | 62.3 | 39.1 | - | - | - | - |
| Qwen-VL-Chat [4] | Qwen-7B | 448 × 448 | 1.4B | 50M | - | - | - | 58.2 | 65.4 | 37.8 | 1487.5 | - | - | 35.9 |
| LLaVA 7B [39] | Vicuna-7B | 336 × 336 | 595K | 158K | 76.3 | 72.2 | 70.1 | 33.5 | 37.0 | 23.8 | 809.6 | 62.8 | 25.5 | - |
| LLaVA-1.5-7B [37] | Vicuna-7B | 336 × 336 | 558K | 665K | 87.3 | 86.1 | 84.2 | 58.6 | 66.1 | 37.3 | 1510.7 | 65.4 | 31.1 | 35.3 |
| LLaVA-1.5-13B [37] | Vicuna-13B | 336 × 336 | 558K | 665K | 87.1 | 86.2 | 84.5 | 61.6 | 68.2 | 42.7 | 1531.3 | 72.5 | 36.1 | 36.4 |
| Direct-LLaVA | Vicuna-7B | 336 × 336 | 558K | 665K | 88.8 | 88.9 | 86.0 | 63.3 | 69.7 | 38.8 | 1524.9 | 69.7 | 32.8 | 36.1 |
| Direct-LLaVA | Qwen-7B | 336 × 336 | 558K | 665K | 89.0 | 88.3 | 87.0 | 64.0 | 70.4 | 40.0 | 1538.1 | 71.0 | 36.0 | 36.9 |
| Direct-LLaVA | LLaMA-8B | 336 × 336 | 558K | 665K | 89.2 | 87.0 | 83.4 | 64.2 | 70.5 | 40.1 | 1555.1 | 71.1 | 36.0 | 37.6 |
| Direct-LLaVA | Vicuna-13B | 336 × 336 | 558K | 665K | **90.5** | **89.2** | **87.8** | **65.6** | **71.5** | **43.3** | **1572.1** | 72.3 | **41.1** | **38.0** |

LLMs with Generative Comprehension Benchmark (SEED-Bench) [22], Comprehensive Evaluation Benchmark of LMM (MME) [59], LLaVA Benchmark in the Wild (LLaVA-Wild) [39], Integrated Capability Benchmark (MM-Vet) [62], and Massive Multi-discipline Multimodal Understanding and Reasoning (MMMU-Val) [63].

## 4.2 Comparision with State-of-the-Art Methods

**Comparison with SoTA methods on Academic-task-oriented Benchmarks.** Table 2 highlights the performance of our approach, evaluated with different LLM versions on various academic-focused benchmarks against previous SoTA methods. The results underscore that our approach surpasses prior benchmarks in accuracy across these tasks. Specifically, Direct-LLaVA, when integrated with Vicuna-13B [10], consistently outperforms the previous SoTA, LLaVA-1.5, which also employs Vicuna-13B. Notable achievements include 77.3% and 65.4% accuracy on ScienceQA and TextVQA benchmarks, respectively, surpassing LLaVA-1.5's results of 71.6% and 61.3%. Additionally, Direct-LLaVA shows considerable accuracy improvements on VQAv2, GQA, and VizWiz, with increases of 1.6%, 2%, and 1.3%.

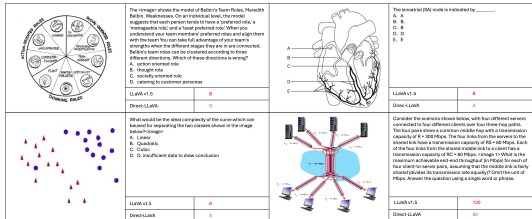

Figure 5: Comparison of Response in Multiple-Choice Questions Between Direct-LLaVA-7B and LLaVA-v1.5-7B on MMMU-Val.

**Comparison with SoTA methods on Instruction-Following LMM Benchmark.** As in Table 3, our method demonstrates superior performance on several instruction-following benchmarks relative to previous approaches. Our model achieves F1 scores of 90.5%, 89.2%, and 87.8% on the POPE benchmark's random, popular, and adversarial settings, surpassing LLaVA-1.5 [37] with Vicuna-13B, which achieved 87.1%, 86.2%, and 84.5%, respectively. Additionally, Direct-LLaVA enhances SEED-Bench performance, achieving an accuracy of 65.5% compared to LLaVA-1.5's 61.6%. Our model also records a score of 1572.1 on the MME benchmark, 41.1% on MM-Vet [62] , and 38% overall accuracy on MMMU-Val [63], underlining its efficacy across complex understanding tasks.

**Visualization.** Fig. 5 illustrates the response of our Direct-LLaVA on MMMU-Val compared to LLaVA v1.5. Our responses are correct and consistent with the information represented by the visual image. In addition, to further illustrate the robustness of our approach, Fig. 6 visualize the conversation about the image between our Direct-LLaVA model and humans with a set of queries. As shown in this conversation, the Direct-LLaVA model can better understand visual information and answer correctly and consistently. Meanwhile, LLaVA tends to produce hallucinated answers produced by LLM that are inconsistent with the images.

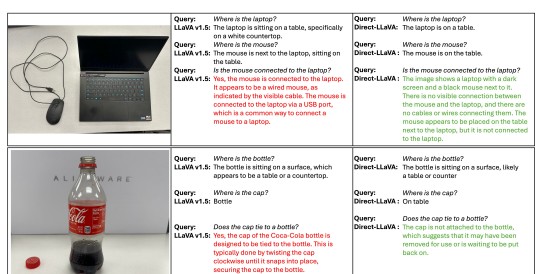

Figure 6: Comparison of Response in Conversation Between Direct-LLaVA-7B and LLaVA-v1.5-7B on In-the-Wild Samples.

## 4.3 Ablation Study

**Effectiveness of Large Language Models.** Table 4 provides an in-depth analysis of how various LLMs impact performance when integrated with different LMMs, i.e., Vicuna-7B, Vicuna-13B, LLaMA-8B [13], and Qwen-7B [3]. In the case of LLaVA, the larger 13B model consistently demonstrates superior results compared to the smaller 7B model across both academic-oriented and instruction-following benchmarks.

Table 4: Effectiveness of Different Language Models.

| Method | LLM | SciQA IMG | Text VQA | POPE rand | POPE pop | POPE adv | MME | SEED-Bench all | SEED-Bench img | SEED-Bench vid |
|---|---|---|---|---|---|---|---|---|---|---|
| LLaVA-v1.5 | Vicuna 7B | 66.8 | 58.2 | 87.3 | 86.1 | 84.2 | 1510.7 | 58.6 | 66.1 | 37.3 |
| LLaVA-v1.5 | Vicuna 13B | 71.6 | 61.3 | 87.1 | 86.2 | 84.5 | 1531.3 | 61.6 | 68.2 | 42.7 |
| Direct-LLaVA | Vicuna 7B | 74.3 | 63.2 | 88.8 | 88.9 | 86.0 | 1524.9 | 63.3 | 69.7 | 38.8 |
| Direct-LLaVA | Qwen | 75.7 | 65.1 | 89.0 | 88.3 | 87.0 | 1538.1 | 64.0 | 70.4 | 40.0 |
| Direct-LLaVA | LLaMA | 75.8 | 64.9 | 89.2 | 87.0 | 83.4 | 1555.1 | 64.2 | 70.5 | 40.1 |
| Direct-LLaVA | Vicuna 13B | **77.3** | **65.4** | **90.5** | **89.2** | **87.8** | **1572.1** | **65.6** | **71.5** | **43.3** |

oriented and instruction-following benchmarks. Although the POPE F1 score shows only a minor increase, other benchmarks exhibit notable improvements with the Vicuna-13B version. This trend holds during the evaluation of these Vicuna versions within Direct-LLaVA. For instance, Direct-LLaVA integrated with the larger Vicuna model increases the performance on ScienceQA and MME from 74.3% to 77.3% and from 1524.9 to 1572.1, respectively. In addition, with a larger model size, LLaMA outperforms compared to Qwen in most benchmarks, with a few exceptions. For example, Qwen slightly surpasses LLaMA with an accuracy of 65.1% compared to LLaMA's 64.9% in TextVQA evaluations. Overall, utilizing Vicuna-13B as the LLM within our Direct-LLaVA framework yields the SoTA performance.

**Effectiveness of Directed Tokens.** We analyze the impact of directed tokens on the performance of our Direct-LLaVA compared to using only visual tokens. As in Table 5, our results indicate that with directed tokens, the model consistently achieves higher scores across both academic-oriented and instruction-following benchmarks, regardless of the language model used. SEED-Bench, for instance, is witnessed a significant boost in both image and video evaluation settings, with the accuracy metric increasing from 68.2% and 42.7% to 71.5% and 43.3%. Furthermore, while the performance of Direct-LLaVA with visual tokens alone is not as strong as with directed tokens, it still outperforms the original LLaVA. Notably, the POPE F1 scores across random, popular, and adversarial split each increase by approximately 2%, alongside a substantial improvement in the MME benchmark, with scores increasing from 1531.3 to 1541.7. This experiment, therefore, confirms that our Direct-LLaVA not only surpasses the baseline LLaVA model on these complex tasks but that the directed token usage further enhances its effectiveness.

**Effectiveness of Shuffle Learning in Pre-Training Phase.** To assess the impact of the shuffling technique, we conduct experiments under three configurations: shuffling images only, shuffling text only, and shuffling both text and images during the pre-training phase. As portrayed in Table 6, integrating directed tokens to address the image ordering reconstruction problem slightly outperforms their use in

Table 5: Effectiveness of Token Selection, i.e., Visual Tokens (**vis**) and Directed Tokens (**drt**) For Reconstructing Image Order Task.

| Method | Token | SciQA IMG | Text VQA | POPE rand | POPE pop | POPE adv | MME | SEED-Bench all | SEED-Bench img | SEED-Bench vid |
|---|---|---|---|---|---|---|---|---|---|---|
| LLaVA-v1.5-7B | | 66.8 | 58.2 | 87.3 | 86.1 | 84.2 | 1510.7 | 58.6 | 66.1 | 37.3 |
| Direct-LLaVA-7B | vis | 71.5 | 61.1 | 88.1 | 87.4 | 85.8 | 1520.4 | 61.1 | 67.1 | 38.5 |
| Direct-LLaVA-7B | drt | **74.3** | **63.2** | **88.8** | **88.9** | **86.0** | **1524.9** | **63.3** | **69.7** | **38.8** |
| LLaVA-v1.5-13B | | 71.6 | 61.3 | 87.1 | 86.2 | 84.5 | 1531.3 | 61.6 | 68.2 | 42.7 |
| Direct-LLaVA-13B | vis | 74.4 | 62.3 | 88.8 | 88.4 | 86.2 | 1541.7 | 62.6 | 67.9 | 42.7 |
| Direct-LLaVA-13B | drt | **77.3** | **65.4** | **90.5** | **89.2** | **87.8** | **1572.1** | **65.6** | **71.5** | **43.3** |

solving text ordering, indicating that shuffling images proves more effective than shuffling text. Furthermore, combining both image and text shuffling during pre-training yields improvements across all benchmarks. Notably, in this setup, the MME score reaches 1519.3, surpassing scores of 1513.5 and 1516.3 achieved with single-modality shuffling. Additionally, our Direct-LLaVA-7B consistently outperforms the prior LLaVA-7B model across all scenarios, confirming that leveraging directed tokens to tackle both text and image shuffling enhances competitive performance overall.

**Effectiveness of Shuffle Learning in Fine-tuning Phase.** We investigate the impact of shuffling images during the fine-tuning stage. As shown in Table 6, incorporating image shuffling leads to notable improvements in performance across both academic-oriented and instruction-following benchmarks. Remarkably, accuracy scores in ScienceQA and TextVQA increase from 71.0% and 61.3% to 72.3% and 62.3%, respectively. Additionally,

Table 6: Effectiveness of Shuffle Learning and Image-to-Response Guided Learning Approaches. Pretraining (Pret.), Finetuning (Fine.), Image (I) and Text (T).

| Method | Pret. T. | Pret. I. | Fine. I. | $\mathcal{L}_{I \to R}$ | SciQA IMG | Text VQA | POPE rand | POPE pop | POPE adv | MME | SEED-Bench all | SEED-Bench img | SEED-Bench vid |
|---|---|---|---|---|---|---|---|---|---|---|---|---|---|
| LLaVA-v1.5-7B | ✗ | ✗ | ✗ | ✗ | 66.8 | 58.2 | 87.3 | 86.1 | 84.2 | 1510.7 | 58.6 | 66.1 | 37.3 |
| LLaVA-v1.5-7B | ✗ | ✗ | ✗ | ✓ | 69.9 | 60.2 | 87.9 | 87.8 | 85.1 | 1518.4 | 59.9 | 67.4 | 38.4 |
| Direct-LLaVA-7B | ✓ | ✗ | ✗ | ✗ | 69.0 | 59.5 | 87.9 | 86.3 | 84.4 | 1513.5 | 60.1 | 65.2 | 41.1 |
| Direct-LLaVA-7B | ✗ | ✓ | ✗ | ✗ | 69.2 | 60.3 | 88.1 | 86.4 | 84.4 | 1516.3 | 60.2 | 65.3 | 41.2 |
| Direct-LLaVA-7B | ✓ | ✓ | ✗ | ✗ | 71.0 | 61.3 | 88.4 | 87.2 | 84.7 | 1519.3 | 61.7 | 66.9 | 42.1 |
| Direct-LLaVA-7B | ✓ | ✓ | ✓ | ✗ | 72.3 | 62.3 | 88.5 | 87.5 | 85.2 | 1521.4 | 62.3 | 67.6 | 42.5 |
| Direct-LLaVA-7B | ✓ | ✓ | ✓ | ✓ | **74.3** | **63.2** | **88.8** | **88.9** | **86.0** | **1524.9** | **63.3** | **69.7** | **38.8** |

the SEED-Bench accuracy for both image and video comprehension tasks shows a 1% improvement. These findings underscore the advantages of applying image shuffling in the fine-tuning process.

**Effectiveness of Image-to-Response Guided Loss.** We evaluate the impact of attention information loss on model performance by comparing outcomes with and without such loss. As in Table 6, the application of Image-to-Response Guided Loss ($\mathcal{L}_{I \rightarrow R}$) boosts the model's performance across various benchmarks. In particular, results for two academic-oriented benchmarks improve from 72.3% and 62.3% to 74.3% and 63.2%, respectively. The POPE F1 Score also increases by 0.3%, 1.4%, and 0.8% on random, popular, and adversarial splits. Additionally, the MME benchmark score climbs by 3.5, while SEED-Bench accuracy advances from 62.3% to 63.3%. This indicates that our loss objective contributes positively to the overall performance.

## 5 Conclusions

Our paper has presented a new shuffling learning approach to improving the robustness and alignment between visual and textual features. In particular, we have introduced two new learning tasks, i.e., reconstructing image order and reconstructing text order, into the pre-training and fine-tuning phases. Our new learning tasks have significantly improved visual understanding and cross-modality alignment of the LMM. To effectively support the reconstructing image order task, we have introduced a new directed token approach to capture both visual and textual features effectively. Then, to further enhance the correlation between visual tokens and the LMM's responses, the new Image-to-Response Guided loss has been introduced. Through intensive experiments on various academic task-oriented and instruction-following LMM benchmarks, our proposed approach has achieved SoTA performance. Our study explores the effectiveness of proposed learning tasks and losses for improving LMM performance under selected hyperparameters and model scales. However, it still has limitations related to objective balancing. Our detailed discussion of limitations is provided in the Appendix.

**Acknowledgment.** This work is partly supported by NSF CAREER (No. 2442295), NSF SCH (No. 2501021), NSF E-RISE (No. 2445877), NSF SBIR Phase 2 (No. 2247237) and USDA/NIFA Award. We also acknowledge the Arkansas High-Performance Computing Center (HPC) for GPU servers.

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

# Appendices

## A Benchmarks

Following the standard benchmarks of LLaVa v1.5, we evaluate our models on two sets of benchmarks, i.e., Academic Task-oriented Benchmarks and Instruction-Following LMM Benchmarks. For the academic task-oriented benchmarks, we adopt five different benchmarks, including Visual Question Answering V2 (VQAv2) [16], Question Answering on Image Scene Graphs (GQA) [18], Answer Visual Questions from People Who Are Blind (VizWiz) [17], Science Question Answering (SciQA-IMG) [41], and Visual Reasoning based on Text in Images (TextVQA) [47]. While VQAv2 and GQA focus on evaluating the visual understanding based on the open-ended short answers, VizWiz evaluates the generalization of the model based on the visual questions raised by impaired people. On the other hand, SciQA-IMG benchmarks will measure the performance of the LMM on scientific questions via multiple-choice questions. The TextVQA benchmark evaluates the capability of models in reading and reasoning about text in images. Meanwhile, we use six Instruction-Following LMM Benchmarks to evaluate our proposed approach, including Polling-based Object Probing Evaluation for Object Hallucination (POPE) [32], Multimodal LLMs with Generative Comprehension Benchmark (SEED-Bench) [22], Comprehensive Evaluation Benchmark of LMM (MME) [59], LLaVA Benchmark in the Wild (LLaVA-Wild) [39], Integrated Capability Benchmark (MM-Vet) [62], and Massive Multi-discipline Multimodal Understanding and Reasoning Benchmark (MMMU-Val) [63]. While the POPE benchmark evaluates the hallucination of the model based on the tree subset, i.e., random (rand), common (pop adv), and adversarial (adv), SEED-Bench measures the generative comprehension of the LMM on both images and videos with multiple-choice questions. For the video evaluation, we adopt the middle frame of the video as a visual input. The MME perception benchmark evaluates visual understanding of the LMM via binary (yes/no) questions. Meanwhile, the LLaVA-Wild and MM-Vet benchmarks measure the capabilities of the LMM in engaging in visual conversations. The MMMU benchmark evaluates LMMs on massive multi-discipline tasks demanding high-level subject knowledge and reasoning.

## B Additional Ablation Study

**Effectiveness of Data Size.** Table 7 presents a comparative analysis of LLaVA-7B and Direct-LLaVA-7B performance across varying data sizes during pre-training and fine-tuning, evaluated on a range of LLM benchmarks. We use the data training size of LLaVA v1 [39] and LLaVA v1.5 [37] for our experiments. Overall, both models demonstrate enhanced outcomes with larger fine-tuning datasets, with Direct-LLaVA consistently surpassing LLaVA. Notably, performance on instruction-following benchmarks, such as POPE, MME, and SEED-Bench, improves substantially as fine-tuning data increases. Specifically, the F1 score on the POPE benchmark rises by approximately 12.5% across different configurations. For the MME benchmark, the scores of LLaVA and Direct-LLaVA increase markedly from 809.6 and 1102.1 to 1510.7 and 1524.9, respectively. The accuracy gain on SEED-Bench is also significant, with larger datasets nearly doubling the models' performance. Furthermore, Direct-LLaVA-7B consistently outperforms LLaVA across both academic-oriented and instruction-following benchmarks. For instance, on academic benchmarks, Direct-LLaVA-7B achieves a 7.5% and 5% higher accuracy in ScienceQA and TextVQA, respectively. These results underscore the robustness of our proposed method.

Table 7: Effectiveness of Pre-training and Fine-tuning Data Size.

| Method | Data Size | | SciQA | Text | POPE | | | MME | SEED-Bench | | |
|---|---|---|---|---|---|---|---|---|---|---|---|
| | Pretrain | Finetune | IMG | VQA | rand | pop | adv | | all | img | vid |
| LLaVA-7B | 595K | 158K | 46.9 | 26.1 | 76.3 | 72.2 | 70.1 | 809.6 | 33.5 | 37.0 | 23.8 |
| Direct-LLaVA-7B | 595K | 158K | **54.6** | **29.3** | **79.5** | **76.2** | **74.3** | **1102.1** | **36.5** | **40.3** | **27.7** |
| LLaVA-v1.5-7B | 558K | 665K | 66.8 | 58.2 | 87.3 | 86.1 | 84.2 | 1510.7 | 58.6 | 66.1 | 37.3 |
| Direct-LLaVA-7B | 558K | 665K | **74.3** | **63.2** | **88.8** | **88.9** | **86.0** | **1524.9** | **63.3** | **69.7** | **38.8** |

**Scalability to Larger Data and Benchmarks.** To illustrate our scalability to larger data and other benchmarks, we conduct experiments on LLaVA-OneVision data [25] with LLaVA and Qwen2.5-

0.5B. We report our results on MME, MMMU, SeedBench-IMG, AI2D [21], and MMBench [40]. As shown in Table 8, when data is scaling up, our proposed approach still maintains its effectiveness and significantly improves the performance of LMM on various benchmarks.

Table 8: Effectiveness of Direct-LLaVA on Large Dataset.

|  | % Samples for Pretext | MME | MMMU | SeedBench-IMG | AI2D | MMBench |
|---|---|---|---|---|---|---|
| LLaVA | - | 1238 | 31.4 | 65.5 | 57.1 | 52.1 |
| Direct-LLaVA | 50% | 1351 | 32.9 | 67.5 | 62.6 | 55.4 |
| Direct-LLaVA | 100% | 1494 | 34.5 | 68.4 | 69.4 | 58.7 |

**Effectiveness of Data In Pretext.** To understand the effectiveness of our proposed approach on the ratio of data used, we conducted an experiment using only 50% of data for our pretext tasks. As shown in Table 8, our approach can improve the performance of the LMMs. The results have further confirmed the effectiveness of our proposed learning approach.

**Visualization of Shuffle Predictions.** We provide the real-world images to illustrate our effectiveness of the shuffling learning mechanism. As shown in Figure 7, for Direct-LLaVA with **cls** (no text in reconstruction), the LMM predicts the image order well but shows noticeable differences from the originals. In contrast, Direct-LLaVA with **drt** (text included) better aligns with the original images since information of language and visuals are well captured in **drt** token. To highlight the impact of text in reconstruction, we altered the description. Although image patches became inconsistent, the images were adapted to match the semantic meaning of the text.

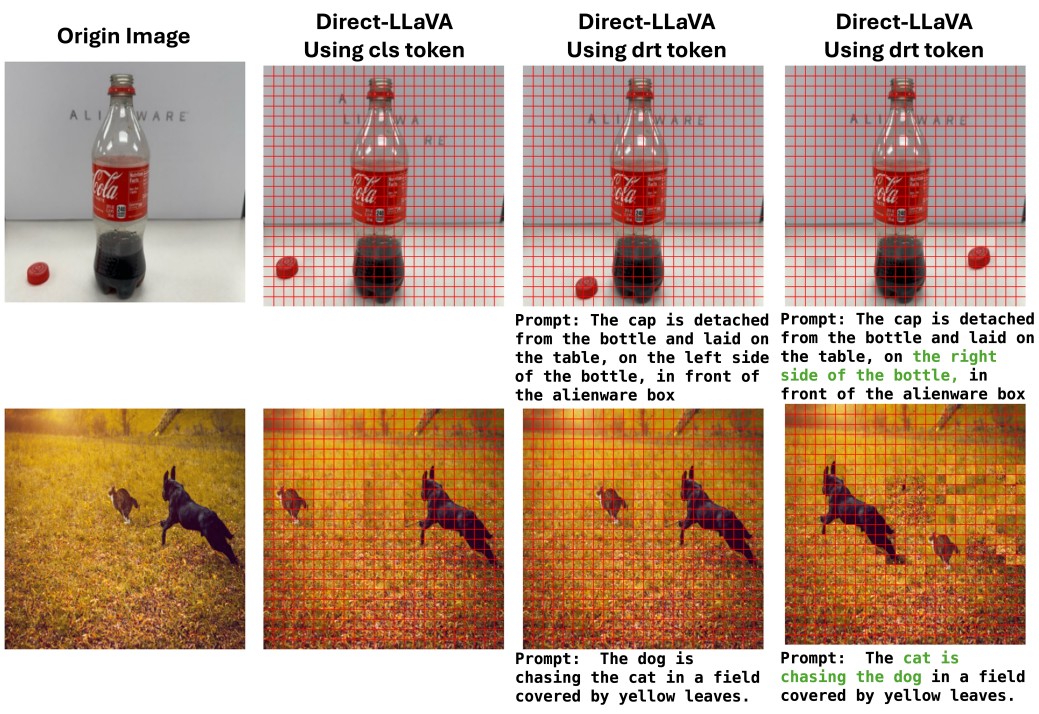

Figure 7: Additional Visualization (Better in $2\times$ zoom for details).

**Ablation study on the placement of the directed token.** The choice to place a single directed token **drt** at the end of the input sequence is to ensure it attends to the full context from both visual and textual modalities under the autoregressive modeling constraint. As shown in Section 3.1, placing the directed token after all other tokens allows it to aggregate multimodal information effectively, which would not be possible if positioned earlier due to the causal attention mask. Our ablation study in Table 9 validates this design: using directed tokens yields consistent performance gains over visual tokens across all benchmarks. In addition, to further justify this design, we conducted

Table 9: Ablation study on difference placement of directed token **drt**

| Placement | Science IMG | TextVQA |
|-----------|-------------|---------|
| Beginning | 70.9 | 61.4 |
| End | **74.3** | **63.2** |

Table 10: Ablation study on number of permutations

| # Permutations | Science IMG | TextVQA |
|----------------|-------------|---------|
| 5000 | 72.5 | 62.1 |
| 10000 | 74.3 | 63.2 |
| 15000 | **74.5** | **64.0** |

an additional experiment comparing placements at the beginning and end of the sequence. Results show that placing the token at the end yields superior performance, supporting our choice.

**Ablations on the number of permutations.** Our selection of 10,000 permutations follows the well-established practice in self-supervised vision learning, particularly works on jigsaw-based tasks [6, 44, 48]. Specifically, we sample permutations to maintain a balanced Hamming distance between them, ensuring sufficient diversity while avoiding overly trivial or degenerate cases. To validate this choice, we conducted an ablation varying the number of permutations. As shown below, increasing from 5,000 to 10,000 leads to improved performance on ScienceQA-IMG and TextVQA, confirming that a richer permutation space helps the model better learn spatial relationships. However, increasing the number to 15,000 leads to only slight improvements, indicating that 10,000 well-chosen permutations already provide sufficient diversity for effective learning, and additional permutations may have limited impact.

**Ablation study on the Image-to-Response Guided loss $\mathcal{L}_{I \to R}$.** To isolate the effect of the Image-to-Response Guided Loss, we conducted an additional experiment by training LLaVA-1.5 with only the next-token prediction loss and the Image-to-Response Guided Loss, excluding our proposed shuffle learning tasks. This setting allows us to directly assess the contribution of the guided loss to visual-textual alignment. As shown in Table 11, our Image-to-Response Guided loss consistently improves performance across all benchmarks, demonstrating its effectiveness in reinforcing visual-textual alignment, yet remains lower than our full Direct-LLaVA model. This result suggests that while the loss improves visual grounding, the Image-to-Response loss alone may not be sufficient for robust multimodal reasoning.

Table 11: Ablation study on Image-to-Response Guided loss

| Model | Science IMG | TextVQA | MME | POPE | | | SEED-Bench | | |
|-------|-------------|---------|-----|------|------|------|------------|------|------|
| | | | | rand | pop | adv | all | img | vid |
| LLaVA | 66.8 | 58.2 | 1510.7 | 87.3 | 86.1 | 84.2 | 58.6 | 66.1 | 37.3 |
| LLaVA with $\mathcal{L}_{I \to R}$ | 69.9 | 60.2 | 1518.4 | 87.9 | 87.8 | 85.1 | 59.9 | 67.4 | 38.4 |
| Direct-LLaVA | **74.3** | **63.2** | **1524.9** | **88.8** | **88.9** | **86.0** | **63.3** | **69.7** | **38.8** |

## C Pseudocode for the training procedure of our framework

The Algorithm 1 elucidates the framework of our proposed Direct-LLaVA during two-stage training process.

## D Discussion of Limitations

Our paper has adopted a specific set of hyper-parameters and learning methods to support our hypothesis. However, our work could contain several limitations. Our work investigated the effectiveness of our proposed learning tasks and losses in improving the LMM's performance. Thus, the investigation of balance weights among learning objectives has not been fully exploited, and we leave this experiment as our future work. Due to computation limitations, our experiments are limited to the standard language model size (i.e., Vicuna 13B, Vicuna 7B, LLaMA3 8B, and Qwen 7B) and data scale (LLaVA v1.5). Nevertheless, we hypothesize that our proposed approaches can generalize to larger-scale language models and data settings due to their fundamental theories.

**Algorithm 1** Training Step in the Two-Stage Training Procedure of Direct-LLaVA

**Input:** Initial model parameters $\theta$, training dataset $\mathcal{D} = (x, p, \{(x_q^t, x_a^t)\}_{t=1}^T)$
**Output:** Updated model parameters $\theta^*$

 1: **for each** iteration $i$ **do**
 2:     $(x, p, \{(x_q^t, x_a^t)\}_{t=1}^{T_i}) \leftarrow \text{GetSample}(\mathcal{D}, i)$
 3:     ▷ **Task 1:** Autoregressive with Cross-Entropy and Guided Attention Losses
 4:     $x_{\text{instruct}} \leftarrow \text{GetInstruction}('\text{ARTask}')$                       ▷ Eqn. (3)
 5:     $x_a \leftarrow p$
 6:     $\mathcal{L}_{\text{CE}} \leftarrow \text{CrossEntropyLoss}(x, x_{\text{instruct}}, x_a)$         ▷ Eqn. (4)
 7:     $\mathcal{L}_{\text{I}\rightarrow\text{R}} \leftarrow \text{ImageToResponseLoss}(x, x_a)$         ▷ Eqn. (12)
 8:     $\theta \leftarrow \theta - \nabla_\theta(\mathcal{L}_{\text{CE}} + \mathcal{L}_{\text{I}\rightarrow\text{R}})$
 9:     ▷ **Task 2:** Image Order Reconstruction
10:     $x_{\text{instruct}}^{\text{image}} \leftarrow \text{GetInstruction}('\text{ImageTask}')$            ▷ Eqn. (7)
11:     $\bar{x} \leftarrow \mathcal{P}(x, k)$                               ▷ Shuffle the image
12:     $\mathcal{L}_{\text{Image-Order}} \leftarrow \text{PredictPermutation}(\bar{x}, x_{\text{instruct}}^{\text{image}})$       ▷ Eqn. (6)
13:     $\theta \leftarrow \theta - \nabla_\theta(\mathcal{L}_{\text{Image-Order}})$
14:     ▷ **Task 3:** Text Order Reconstruction (Pre-training Only)
15:     $x_{\text{instruct}}^{\text{text}} \leftarrow \text{GetInstruction}('\text{TextTask}')$              ▷ Eqn. (8)
16:     $x_a^{\text{text}} \leftarrow \texttt{concat}(p, \mathbf{drt})$
17:     $\mathcal{L}_{\text{Text-Order}} \leftarrow \text{ReorderPrediction}(x, x_{\text{instruct}}^{\text{text}}, x_a^{\text{text}})$     ▷ Eqn. (9)
18:     $\theta \leftarrow \theta - \nabla_\theta(\mathcal{L}_{\text{Text-Order}})$
19: **end for**

