# OpenReview forum: "Directed-Tokens: A Robust Multi-Modality Alignment Approach to Large Language-Vision Models"
_NeurIPS.cc/2025/Conference — NeurIPS 2025 poster_

### Official Review · Reviewer_5PqY · 2025-06-23

**Clarity:** 2
**Significance:** 4
**Originality:** 1
**Rating:** 4
**Confidence:** 4

**Summary:**

This paper aims at improving the vision-language alignment of Large Multimodal Models (LMMs). It starts by ascertaining that the popular LMM LLaVA 1.5 pays little attention to image inputs, as its VQA performance is just moderately decreased if the visual input is removed. Inspired by the jigsaw puzzle pre-training scheme, the paper proposes to add three different losses in the pre-training and visual instruction tuning stages of LLaVA: the first loss tasks the model to reconstruct the order of the input image where the patches have been permuted; similarly, the second loss tasks the model to reconstruct the order of the permuted response. This is only applied during pre-training; finally, the third Image-to-Response Guided Learning loss encourages the model to pay more attention to visual tokens. The reconstruction of the image in the right order is feasible thanks to a set of Directed Tokens that are additionally fed to the LMM and whose representation at the output layer serves to predict the index of the k-th image patch.
Experiments are done on the same dataset employed by LLaVA and demonstrate consistent improvements over LLaVA 1.5.

**Questions:**

See **WEAKNESSES** above.

**Ethical Concerns:**

["NO or VERY MINOR ethics concerns only"]

**Final Justification:**

After reading the rebuttal, my concerns have been solved, and I think that the paper now meets the acceptance bar.

**Limitations:**

yes

**Paper Formatting Concerns:**

Tables font size is small, making it difficult to read numbers.

**Quality:**

3

**Strengths And Weaknesses:**

**STRENGTHS:**

Strong results. While the idea of shuffle learning is not new, its application to image and text, along with the Image-to-Response loss, brings consistent and solid improvements over LLaVA across a multitude of benchmarks, given the same training data. The effectiveness of the proposed method remains persistent even when varying the Large Language Models.


**WEAKNESSES:**

1. Ablation studies in Table 6 lack a fundamental experiment to isolate the role of the Image-to-Response Guided Loss. The last row of the table does show its effect when paired with the shuffle learning objectives, but we do not know what would be the results of training LLaVA 1.5 only with the Image-to-Response Guided Loss, in addition to the usual next token prediction loss. Please add this during the rebuttal.

2. Computational cost. The authors experiment with 32 A100 GPUs, while LLaVA 1.5 only requires 8 A100 GPUs (assuming 80GB of VRAM for each A100). I imagine this is because each training sample must be forwarded to the model three times: one for the next token prediction with original image and text, one for image shuffle learning with the permuted image, and one for text shuffle learning with the permuted text. Consequently, to keep the same batch size as LLaVA, it is necessary to increase GPUs. One benefit of this method is to improve LLaVA without having access to new training data. Yet, computational resources are important too. It would be good to include the results of Direct-LLaVA trained with the same amount of GPUs as LLaVA 1.5.

3. The choice of the shuffling strategy (L276-287) is theoretically justified, but it lacks empirical ablation. What does it happen if we just use purely random permutations?

4. Presentation. I feel that figures, and tables in particular, have a small font size, making it difficult to read numbers. Moreover, I do not see the point of including Table 4, as it reports the very same results as Tables 2 and 3.

5. It would be interesting to repeat the experiment from Table 1 with Direct-LLaVA too.

---

> ### Author Rebuttal · Authors · 2025-07-30
>
> Dear Reviewer 5PqY,
>
>
> We sincerely appreciate your constructive feedback. We are pleased that you found our method to yield robust and consistent improvements across benchmarks and LLMs. We would like to address your points in detail below.
>
>
> [Q1] **Ablation studies in Table 6 lack a fundamental experiment to isolate the role of the Image-to-Response Guided Loss.**
>
>
> [A1] To isolate the effect of the Image-to-Response Guided Loss, we conducted an additional experiment by training LLaVA-1.5 with only the next-token prediction loss and the Image-to-Response Guided Loss, excluding our proposed shuffle learning tasks. This setting allows us to directly assess the contribution of the guided loss to visual-textual alignment. The resulting performance on ScienceQA-IMG and TextVQA shows improvement over the LLaVA-1.5 baseline, but remains lower than our full Direct-LLaVA model. This result suggests that while the loss improves visual grounding, the Image-to-Response loss alone may not be sufficient for robust multimodal reasoning.
>
>
> |                        | Science IMG | TextVQA |
> | ---------------------- | ----------- | ------- |
> | LLaVA                  | 66.8        | 58.2    |
> | LLaVA with I-to-R Loss | 69.9        | 60.2    |
> | Direct-LLaVA           | **74.3**        | **63.2**    |
>
>
> [Q2] **Computational cost: Direct-LLaVA trained with the same number of GPUs as LLaVA 1.5.**
>
>
> [A2] We clarify that our experiments were conducted using A100 GPUs with 40GB of VRAM, not 80GB. We employed 32 GPUs primarily to accelerate training and support the breadth of ablation studies, not to increase batch size or model capacity. Importantly, despite the additional compute, we strictly maintained the same training hyperparameters as LLaVA-1.5, including batch size, learning rate, optimizer, and number of epochs, to ensure a fair and controlled comparison.
>
>
>
>
> [Q3] **The use of purely random permutations**
>
>
> [A3] In the early development phase, we have already experimented with purely random permutations sampled from the full permutation space. However, the model failed to converge under this setting. This is primarily because visual random permutations are extremely hard to predict, especially when the image patches are rearranged into highly disordered configurations with no spatial continuity. This randomness creates an enormous and unstructured search space for the permutation prediction task, resulting in unstable and ineffective learning objectives. Moreover, many of these permutations are either too trivial or too chaotic, providing inconsistent or uninformative supervision. To address this, we adopted a more structured strategy by sampling 10,000 permutations with controlled Hamming distances, following established practices in self-supervised learning (e.g., [1, 2, 3]). This design ensures a more learnable yet sufficiently diverse permutation space, enabling better training stability and performance. To validate this choice, we conducted an ablation varying the number of permutations. As shown below, increasing from 5,000 to 10,000 leads to improved performance on ScienceQA-IMG and TextVQA, confirming that a richer permutation space helps the model better learn spatial relationships. However, increasing the number to 15,000 leads to only slight improvements, indicating that 10,000 well-chosen permutations already provide sufficient diversity for effective learning, and additional permutations may have limited impact.
>
>
> | # Permutations      | Science IMG | TextVQA |
> | ----- | ----------- | ------- |
> | 5000  | 72.5        | 62.1    |
> | 10000 | 74.3        | 63.2    |
> | 15000 | **74.5**        | **64.0**    |
>
>
> [Q4] **Presentation of some tables and figures**
>
>
> [A4] We will revise the figures and tables to ensure that all numerical values are clearly legible in the camera-ready version. Regarding Table 4, its purpose is to highlight the impact of different large language models (LLMs) under a unified setting (Direct-LLaVA and LLaVA baselines) to isolate the effect of the LLM backbone. While some values overlap with Tables 2 and 3, Table 4 provides a focused comparison across models using the same architecture and training setup, which is not the emphasis of Tables 2 and 3.
>
>
> [Q5] **It would be interesting to repeat the experiment from Table 1 with Direct-LLaVA too.**
>
>
> [A5] We conducted the same experiment as in Table 1 by replacing the visual input with a black image and evaluating Direct-LLaVA. However, unlike LLaVA, our model is explicitly trained to depend on meaningful visual input through image order reconstruction and Image-to-Response Guided Loss. As a result, when presented with blank images, Direct-LLaVA consistently refuses to generate an answer or outputs an empty response, leading to 0\% accuracy across test samples. This behavior reflects the model's learned reliance on visual grounding and its sensitivity to missing or meaningless visual inputs. For this reason, we report the result as N/A to indicate that the model does not produce valid outputs under such conditions, rather than interpreting the result as typical task performance. We will clarify this behavior in the revised version.
>
>
> | LMM         | Image  | SciQA-IMG | MMMU-Val |
> | ----------- | ------ | --------- | -------- |
> | LLaVA v1.5  | Origin | 66.8      | 35.3     |
> | LLaVA v1.5  | Black  | 64.1      | 32.4     |
> | DirectLLaVA | Origin | **74.3**      | **36.1**     |
> | DirectLLaVA | Black  | N/A       | N/A      |
>
>
>
>
> **References**
>
>
> [1] Carlucci, et al. Domain generalization by solving jigsaw puzzles. CVPR, 2019.
>
>
> [2] Noroozi, et al. Unsupervised learning of visual representations by solving jigsaw puzzles. ECCV. 2016.
>
>
> [3] Truong, et al. DirecFormer: A Directed Attention in Transformer Approach to Robust Action Recognition. CVPR, 2022.

---

> > ### Comment · Reviewer_5PqY · 2025-08-02
> >
> > I thank the authors for their response and the time spent running new experiments.
> >
> > I am satisfied with the response about Q2, Q3, Q4, and Q5. I particularly cared about the computational cost of the proposed method. Since your GPUs have half the memory of those used in LLaVA, I think that it is acceptable, given the overall improvements over LLaVA. My first understanding of Direct-LLaVA was that each sample during pre-training/fine-tuning was repeated to compute the different losses, but I was wrong. I encourage the authors to clarify this point in the paper, as they did in the response to Reviewer kMkc (Q5).
> >
> > Speaking about my Q1, I encourage the authors to be consistent with Table 6 by evaluating *LLaVA with I-to-R Loss* also on POPE, MME, and SEED-Bench. As you have already trained the model, the evaluation should be relatively cheap to run.

---

> > > ### Author Response · Authors · 2025-08-02
> > >
> > > Dear Reviewer 5PqY,
> > >
> > >
> > > We sincerely thank the reviewer for the detailed and constructive feedback, and we’re glad to hear that our answers have addressed your concerns in Q2-Q5. As a follow-up to Q1, we agree that a more complete evaluation of the Image-to-Response Guided Loss is important for consistency with Table 6. We have now conducted the additional evaluations on POPE, MME, and SEED-Bench.
> > >
> > >
> > > |                        | Science IMG | TextVQA |      | POPE |      | MME    |      | SEED-Bench |      |
> > > |------------------------|-------------|---------|------|------|------|--------|------|------------|------|
> > > |                        |             |         | rand | pop  | adv  |        | all  | img        | vid  |
> > > | LLaVA                  | 66.8        | 58.2    | 87.3 | 86.1 | 84.2 | 1510.7 | 58.6 | 66.1       | 37.3 |
> > > | LLaVA with I-to-R Loss | 69.9        | 60.2    | 87.9 | 87.8 | 85.1 | 1518.4 | 59.9 | 67.4       | 38.4 |
> > > | Direct-LLaVA           | **74.3**        | **63.2**    | **88.8** | **88.9** | **86.0** | **1524.9** | **63.3** | **69.7**       | **38.8** |
> > >
> > >
> > > As shown, our Image-to-Response Guided loss consistently improves performance across all benchmarks, demonstrating its effectiveness in reinforcing visual-textual alignment, even when used independently. Our results further validate the effectiveness of our proposed image-to-response guided loss. We will include this ablation study in our revised version.
> > >
> > >
> > > Please let us know if you have any other questions.

---

> > > > ### Comment · Reviewer_5PqY · 2025-08-05
> > > >
> > > > I appreciate the authors' timely response. I will raise my score to borderline accept, as the experimental validation of Directed-Tokens is solid.

---

> > > > > ### Author Response · Authors · 2025-08-05
> > > > >
> > > > > Dear Reviewer 5PqY,
> > > > >
> > > > > Thank you very much for your positive feedback and for raising your rating. Your insightful feedback has helped us to improve the paper.
> > > > >
> > > > > Thank you very much,
> > > > >
> > > > > Authors

---

### Official Review · Reviewer_kMkc · 2025-07-03

**Clarity:** 2
**Significance:** 3
**Originality:** 2
**Rating:** 4
**Confidence:** 4

**Summary:**

This paper proposes Directed Tokens, a simple and effective approach to enhance multimodal alignment in LMMs through shuffle learning tasks. Specifically, it introduces two ordering tasks for the image and text, respectively, with a directed token to strengthen multimodal understanding alignment. To better utilize visual information during response generation, an Image-to-Response Guided Loss is further designed. Experiments show that the proposed method achieves superior results on multiple benchmarks. However, I have some concerns about this paper.

**Questions:**

Refer to Weaknesses

**Ethical Concerns:**

["NO or VERY MINOR ethics concerns only"]

**Final Justification:**

The response has addressed my main concerns about the manuscript, and the alignment of vision and language is of great significance in the MLLM field. Therefore, I adjust my rating to borderline accept, acknowledging the potential contribution of this work to the community.

**Limitations:**

Yes

**Quality:**

3

**Strengths And Weaknesses:**

**Positive points**
1. The paper introduces a novel shuffle learning paradigm coupled with directed-token mechanisms, effectively enhancing visual-textual alignment in LMMs by forcing models to reconstruct correct modality orders during both pre-training and fine-tuning phases.
2. The proposed Image-to-Response Guided loss explicitly enhances visual feature utilization during inference through attention-layer optimization, significantly reducing language-preference bias while maintaining the original autoregressive modeling framework.

**Negative points**
1. The core idea of this paper is simple and intuitive, but the overall writing quality appears rushed and needs further polishing.
a)Several figures are currently placed in narrow side columns with reduced size. Given the single-column template, reformatting them as full-width figures would improve clarity. In addition, the visual design of some figures and tables looks somewhat rough and could be refined for better presentation quality.
b)Figure and table references are currently written in plain text rather than using cross-referencing.
c)There are also several other issues in equations factual correctness. Please refer to the Minor issues for details.
2. The Related Work section is overly generic regarding large multimodal models. It would benefit from a more structured discussion of representative LMM alignment paradigms, such as cross-attention (Flamingo, CogVLM[A]), Q-former (BLIP-2, MiniGPT-4), and MLP (LLaVA, Qwen2-VL[B]), as well as recent research on auxiliary task design[B] and attention mechanism[C] for improving vision-language alignment.
3. The image permutation reconstruction task lacks sufficient implementation detail. The paper would benefit from a clearer explanation of how the permutation index $k$ is represented and predicted, including the design of the order projection layer and related hyperparameters.
4. The design of the permutation space appears somewhat heuristic. While selecting 10,000 permutations from a large combinatorial space is effective, it would be helpful to include ablations on the number of permutations and patch size.
5. If I understand correctly, the two shuffle learning tasks are designed to be independent, with each forward-backward pass performing either image order reconstruction or text order reconstruction. However, in Equation 13, these two loss terms appear simultaneously, which seems inconsistent with this setup and introduces ambiguity.
6. The paper could benefit from an analysis of failure cases or potential trade-offs. For example, the Image-to-Response Guided Loss may lead the model to over-rely on visual cues, possibly affecting its instruction-following or text-only capabilities.
7. Some experimental results (rows 4, 5, and 6) in Table 3 appear incomplete or too limited in scope. These entries could either be expanded with additional metrics or removed if full results are unavailable.


**Minor Issues:**
1. In Eq. 2, the notation for the image input is denoted as $x$ initially and later as $I$, which is inconsistent.
2. In Eq. 8, $[x, p, \overline{p}]$ should be $[x, q, \overline{p}]$.
3. Eq.4 appears to be redundant with Eq.2.
4. The symbol $L$ is used to represent both the number of layers (in Eq. 12) and the sequence length (in Eq. 2), which may cause confusion.
5. In line 133, Page 4, the phrase “two stages of the instruction-tuning procedure” is inaccurate, as instruction-tuning itself is the second stage.
6. In line 236, Page 6, the expression $x_a^T = [x_a^T, \mathbf{drt}]$ seems problematic due to self-referencing on both sides of the equation.
7. In line 256, Page 6, “the attention score from the visual token $v$ to the response token $r$”may cause confusion, as it suggests the visual token attends to the response token, which contradicts the autoregressive nature of decoding.


[A] Cogvlm: Visual expert for pretrained language models. NeurIPS 2024.
[B] Qwen2-vl: Enhancing vision-language model's perception of the world at any resolution. Arxiv 2024.
[C] Align before fuse: Vision and language representation learning with momentum distillation. NeurIPS 2021.
[D] Visual grounding via accumulated attention. CVPR 2018.

---

> ### Author Rebuttal · Authors · 2025-07-30
>
> Dear Reviewer kMkc,
>
>
> We sincerely appreciate your thoughtful and encouraging feedback. We are pleased that you appreciate the novelty of our modality-shuffling framework and the effectiveness of our guided loss in improving visual-text alignment and reducing language bias. Your comments are deeply valued, and we address your specific points below.
>
>
> [Q1] **The core idea of this paper is simple and intuitive, but the overall writing quality appears rushed and needs further polishing.**
>
>
> [A1] Thank you for the constructive feedback. In the revised version, we will carefully reformat all figures and tables as suggested to ensure better readability. We appreciate the reviewer’s attention to these details and will ensure a polished revision.
>
>
> [Q2] **The Related Work section is overly generic regarding large multimodal models.**
>
>
> [A2] We thank the reviewer for the valuable suggestions [4, 5, 6, 7]. In the revised version, we have incorporated a more structured discussion of alignment paradigms in LMMs, including cross-attention mechanisms (e.g., Flamingo, CogVLM), Q-former-based alignment (e.g., BLIP-2, MiniGPT-4), and MLP-based projections (e.g., LLaVA, Qwen2-VL). These additions help position our work more clearly within the broader landscape of multimodal learning.
>
>
> [Q3] **The image permutation reconstruction task lacks sufficient implementation detail.**
>
>
> [A3] We have detailed our implementation in L163–165. We denote the permutation index as $k$, which represents the specific ordering of image patches applied via a permutation function $P(x, k)$. To predict $k$, we introduce a visual order projection head, implemented as a linear layer applied to the last hidden-state representation of the directed token. The model learns to regress the permutation index $k$ from this representation.
>
>
> [Q4] **Ablations on the number of permutations and patch size.**
>
>
> [A4] Our selection of 10,000 permutations follows the well-established practice in self-supervised vision learning, particularly works on jigsaw-based tasks [1, 2, 3]. Specifically, we sample permutations to maintain a balanced Hamming distance between them, ensuring sufficient diversity while avoiding overly trivial or degenerate cases. To validate this choice, we conducted an ablation varying the number of permutations. As shown below, increasing from 5,000 to 10,000 leads to improved performance on ScienceQA-IMG and TextVQA, confirming that a richer permutation space helps the model better learn spatial relationships. However, increasing the number to 15,000 leads to only slight improvements, indicating that 10,000 well-chosen permutations already provide sufficient diversity for effective learning, and additional permutations may have limited impact.
>
>
> | # Permutations      | Science IMG | TextVQA |
> | ----- | ----------- | ------- |
> | 5000  | 72.5        | 62.1    |
> | 10000 | 74.3        | 63.2    |
> | 15000 | **74.5**        | **64.0**    |
>
>
> [Q5] **Two loss terms appearing simultaneously in Equation 13 introduce ambiguity.**
>
>
> [A5] Eqn (13) is not ambiguous. We clarify that although Eqn. (13) presents the three all loss components in a single expression for conciseness, they are not applied simultaneously within a single forward-backward pass. As clarified in Algorithm 1 (included in the revised version), each training sample is assigned to only one task per step: either autoregressive prediction, image order reconstruction, or text order reconstruction. These tasks are handled in independent forward-backward passes, and the corresponding loss is computed based on the selected task. Eqn. (13) is intended to summarize the overall training objective across the dataset, not a per-step computation. We will update the text surrounding Eqn. (13) to better reflect this and avoid potential confusion.
>
>
> [Q6] **Analysis of failure cases or potential trade-offs.**
>
>
> [A6] We conducted an additional experiment using LLaVA trained with only the Image-to-Response Loss, without incorporating our shuffle learning tasks. The resulting performance on SciQA-IMG and TextVQA is lower than that of our full Direct-LLaVA model, but still demonstrates performance improvement over the LLaVA baseline. This result suggests that while the loss improves visual grounding, the Image-to-Response loss alone may not be sufficient for robust multimodal reasoning.
>
>
> |                        | Science IMG | TextVQA |
> | ---------------------- | ----------- | ------- |
> | LLaVA                  | 66.8        | 58.2    |
> | LLaVA with I-to-R Loss | 69.9        | 60.2    |
> | Direct-LLaVA           | **74.3**        | **63.2**    |
>
>
> [Q7] **Some experimental results in Table 3 appear incomplete or too limited in scope.**
>
>
> [A7] We acknowledge that rows 4, 5, and 6 in Table 3 appear limited in scope due to missing metrics. These entries correspond to baseline models whose full results are not publicly disclosed in the original papers; thus, we have included the available results for completeness and reference. In the revised version, we will clarify this in the table caption and main text. We are open to removing these rows if the limited reporting causes confusion or detracts from clarity.
>
>
> [Q8] **Minor issues**
>
>
> [A8] We would like to thank the reviewer for the suggestion. We have revised and updated the minor issues to improve the presentation of our paper.
>
>
>
>
>
>
> **References**
>
>
> [1] Carlucci, et al. Domain generalization by solving jigsaw puzzles. CVPR, 2019.
>
>
> [2] Noroozi, et al. Unsupervised learning of visual representations by solving jigsaw puzzles. ECCV. 2016.
>
>
> [3] Truong, et al. DirecFormer: A Directed Attention in Transformer Approach to Robust Action Recognition. CVPR, 2022.
>
> [4] Wang et al. Cogvlm: Visual expert for pretrained language models. NeurIPS, 2024.
>
> [5] Wang et al. Qwen2-vl: Enhancing vision-language model's perception of the world at any resolution. arXiv, 2024.
>
> [6] Li et al. Align before fuse: Vision and language representation learning with momentum distillation. NeurIPS, 2021.
>
> [7] Deng et al. Visual grounding via accumulated attention. CVPR, 2018.

---

### Official Review · Reviewer_Xcbu · 2025-07-03

**Clarity:** 3
**Significance:** 3
**Originality:** 3
**Rating:** 4
**Confidence:** 4

**Summary:**

This paper proposes Direct-LLaVA, a framework designed to improve the alignment between visual and textual modalities in large multimodal models (LMMs). The authors introduce two novel shuffle learning tasks: image order reconstruction and text order reconstruction, to be used during both pre-training and fine-tuning. The key contribution is the directed token, which is learnable and is appended to the input sequence to capture cross-modal information. The paper also introduces an Image-to-Response Guided Loss to enforce attention from visual inputs to textual outputs. Experimental results demonstrate state-of-the-art performance across multiple academic and instruction-following benchmarks.

**Questions:**

Questiones:

1. I suggest the authors to reformat the paper, and put some experiments into the appendix.

2. It would be better to have a complete algorithm box.

3. How is the balance among multiple losses (CE, image/text order, and attention-guided loss) tuned? Are these loss weights fixed or adaptively adjusted during training?

**Ethical Concerns:**

["NO or VERY MINOR ethics concerns only"]

**Final Justification:**

My concerns have been well addressed, and I will recommend this paper as "accept."

**Limitations:**

Yes, but the limitation is not sufficient. And it is attached as the supplementary, very difficult to find.

**Paper Formatting Concerns:**

Need to reorganize some tables and figures.

**Quality:**

3

**Strengths And Weaknesses:**

Pros:
1. The proposed “Directed Token” provides a simple yet elegant solution to cross-modal information integration under autoregressive modeling. The token captures both visual and textual cues more effectively than traditional visual tokens, as confirmed through comprehensive ablation studies.

2. This paper is well motivated. The authors demonstrate that existing models like LLaVA often ignore visual inputs, producing nearly identical outputs even with blank images.

3. The model achieves state-of-the-art results across a wide range of academic and instruction-following benchmarks, including VQAv2, GQA, SciQA-IMG, TextVQA, POPE, MMMU, and MME.

Cons:

1. The paper's structure and formatting can be improved. There is no need to put everything in the main context, it would be easier to follow if the authors put some content as the appendix. For example, Table 2 and Figure 5 are too small to be well understood.

2. While the Directed Token is empirically effective, the paper lacks deeper theoretical justification or analysis regarding why it facilitates better cross-modal alignment. Its design is intuitive, but the underlying learning dynamics remain unexplored.

---

> ### Author Rebuttal · Authors · 2025-07-30
>
> Dear Reviewer Xcbu,
>
>
> We sincerely appreciate your thoughtful and encouraging feedback. We are pleased that you found our proposed method a simple yet effective approach to cross-modal integration, with strong motivation and compelling empirical support. Your recognition of our experimental rigor and comprehensive benchmarking is truly appreciated. We value your comments and would like to respond to your points as follows.
>
>
> [Q1] **The paper's structure and formatting can be improved.**
>
>
> [A1] We agree that some structural improvements could enhance readability. In the revised version, we will restructure our paper as your suggestion to improve the presentation of the paper.
>
>
> [Q2] **The paper lacks deeper theoretical justification or analysis regarding why the Directed Token facilitates better cross-modal alignment.**
>
>
> [A2] We thank the reviewer for the thoughtful comment. The directed token is theoretically motivated by the causal attention constraint in autoregressive LMMs, where earlier tokens (e.g., visual inputs) cannot attend to later ones (e.g., text). By placing a learnable token at the end of the sequence, it has full access to both modalities and functions as a global integrator of cross-modal information. From an information-theoretic view, it acts as a bottleneck token, summarizing joint semantics for downstream tasks. This design aligns with our experimental findings. As shown in Table 5, using the directed token significantly improves performance over standard visual tokens.
>
>
> [Q3] **It would be better to have a complete algorithm box.**
>
>
> [A3] We appreciate your suggestion. The subsequent section elucidates the algorithm employed during the two-stage training process.
>
>
> ### **Algorithm: Training Step in the Two-Stage Training Procedure of Direct-LLaVA**
>
>
> **Input**:
> - Initial model parameters: $\theta$
> - Training dataset $\mathcal{D} = \{(x, p, \\{(x_q^t, x_a^t)\\}_{t=1}^T)\}$
> - Number of training iterations: $N$
>
>
> **Output**:
> - Updated model parameters: $\theta^*$
>
>
> ---
>
>
> **for** *each iteration* **do**
> 1. Sample a training instance $(x, p, \\{(x_q^t, x_a^t)\\}_{t=1}^T)$ from $\mathcal{D}$
>
>
> 2. **Autoregressive Task with Cross-Entropy and Guided Attention Loss**
>   a. Construct instruction input:
>   &nbsp;&nbsp;&nbsp;&nbsp;$x\_{\text{instruct}} \leftarrow \text{GetInstruction}(\text{`ARTask'})$ (Eq. 3)
>   &nbsp;&nbsp;&nbsp;&nbsp;$x_a \leftarrow p$
>   b. Compute losses:
>   &nbsp;&nbsp;&nbsp;&nbsp;$\mathcal{L}\_{\text{CE}} \leftarrow \text{CrossEntropy}(x, x\_{\text{instruct}}, x\_a)$ (Eq. 4)
>   &nbsp;&nbsp;&nbsp;&nbsp;$\mathcal{L}\_{\text{I→R}} \leftarrow \text{ImageToResponseLoss}(x, x\_a)$ (Eq. 12)
>   c. Update model:
>   &nbsp;&nbsp;&nbsp;&nbsp;$\theta \leftarrow \theta - \nabla\_\theta (\mathcal{L}\_{\text{CE}} + \mathcal{L}\_{\text{I→R}})$
>
>
> 3. **Image Order Reconstruction Task**
>   a. Construct instruction:
>   &nbsp;&nbsp;&nbsp;&nbsp;$x\_{\text{instruct}}^{\text{image}} \leftarrow \text{GetInstruction}(\text{`ImageTask'})$ (Eq. 7)
>   b. Shuffle image:
>   &nbsp;&nbsp;&nbsp;&nbsp;$\bar{x} \leftarrow \mathcal{P}(x, k)$
>   c. Compute loss:
>   &nbsp;&nbsp;&nbsp;&nbsp;$\mathcal{L}\_{\text{Image-Order}} \leftarrow \text{PredictPermutation}(\bar{x}, x\_{\text{instruct}}^{\text{image}})$ (Eq. 6)
>   d. Update model:
>   &nbsp;&nbsp;&nbsp;&nbsp;$\theta \leftarrow \theta - \nabla\_\theta (\mathcal{L}\_{\text{Image-Order}})$
>
>
> 4. **Text Order Reconstruction Task (Pre-training Only)**
>   a. Construct instruction:
>   &nbsp;&nbsp;&nbsp;&nbsp;$x\_{\text{instruct}}^{\text{text}} \leftarrow \text{GetInstruction}(\text{`TextTask'})$ (Eq. 8)
>   b. Construct answer:
>   &nbsp;&nbsp;&nbsp;&nbsp;$x\_a^{\text{text}} \leftarrow \texttt{concat}(p, \mathbf{drt})$
>   c. Compute loss:
>   &nbsp;&nbsp;&nbsp;&nbsp;$\mathcal{L}\_{\text{Text-Order}} \leftarrow \text{ReorderPrediction}(x, x\_{\text{instruct}}^{\text{text}}, x\_a^{\text{text}})$ (Eq. 9)
>   d. Update model:
>   &nbsp;&nbsp;&nbsp;&nbsp;$\theta \leftarrow \theta - \nabla\_\theta (\mathcal{L}\_{\text{Text-Order}})$
>
>
> **end for**
>
> [Q4] **How is the balance among multiple losses tuned? Are these loss weights fixed or adaptively adjusted during training?**
>
>
> [A4] In our current implementation, we assign equal weighting to all loss components (cross-entropy, image/text order reconstruction, and attention-guided loss) for simplicity and fairness in evaluating their contributions. As noted in our limitations, we have not yet explored loss weighting strategies. Despite this, our model still achieves consistent performance gains across all benchmarks (Tables 2–3), indicating the robustness of the proposed objectives even under uniform weighting. We reserve the investigation of loss weighting for future work.

---

### Official Review · Reviewer_oM4U · 2025-07-03

**Clarity:** 3
**Significance:** 2
**Originality:** 3
**Rating:** 4
**Confidence:** 3

**Summary:**

This paper proposes Direct-LLaVA, a new method that introduces directed tokens and shuffle learning objectives (reconstructing image and text order) into the training of vision-language models (LMMs) to improve multimodal alignment and robustness. The authors further incorporate an Image-to-Response Guided Loss to strengthen the influence of visual inputs in textual outputs. Extensive experiments on academic and instruction-following benchmarks demonstrate consistent performance gains over LLaVA-1.5 and other baselines.

**Questions:**

see weakness

**Ethical Concerns:**

["NO or VERY MINOR ethics concerns only"]

**Final Justification:**

Thank for authors' rebuttal, which addressed my concerns. I will maintain my positive attitude and score.

**Limitations:**

Limitations such as scale constraints and lack of hyperparameter sensitivity exploration are briefly discussed in Appendix C, though more analysis could still be provided.

**Paper Formatting Concerns:**

Conforms to NeurIPS formatting. Figures and tables are legible.

**Quality:**

2

**Strengths And Weaknesses:**

### Strengths

- Introduces two new learning objectives—reconstructing image order and text order—to directly improve cross-modal alignment and reduce over-reliance on the language modality.
- Proposes a novel “directed token” mechanism to aggregate visual and textual information for permutation prediction, showing consistent improvements over baseline designs.
- Achieves SoTA results across multiple benchmarks (e.g., VQAv2, GQA, VizWiz, SciQA-IMG, POPE, SEED-Bench, MMMU-Val) under different LLM backbones (Vicuna, LLaMA, Qwen).
- Presents thorough ablation studies on model components (token types, shuffle strategies, LLMs), demonstrating robustness and effectiveness across different configurations.
- Provides clear implementation details and fair comparisons using consistent training data, model sizes, and evaluation protocols.

### Weaknesses

- The design choice of using a single directed token at the end of the sequence lacks deeper analysis; alternative placements or token designs are not explored.
- There is a lack of analysis of failure cases where multimodal misalignment still occurs; visualizations are limited and do not systematically reveal limitations.
- The shuffle learning tasks are somewhat synthetic and may not fully represent the types of semantic misalignment encountered in real-world scenarios.
- The main components (e.g., permutation prediction, guided attention, special tokens) are individually incremental, and the novelty lies more in the combination than in fundamental algorithmic innovation.

---

> ### Author Rebuttal · Authors · 2025-07-30
>
> Dear Reviewer oM4U,
>
>
> We sincerely appreciate your positive feedback. We are glad you appreciated our novel learning objectives and directed token design, and the robustness of our results across different benchmarks. We are also grateful that the clarity of our methodology and thorough ablation studies were particularly notable. Your comments are very much appreciated, and we would like to address your points as follows.
>
>
> [Q1] **Alternative placements of the directed token**
>
>
> [A1] We chose to place a single directed token at the end of the input sequence to ensure it attends to the full context from both visual and textual modalities under the autoregressive modeling constraint. As shown in Section 3.1, placing the directed token after all other tokens allows it to aggregate multimodal information effectively, which would not be possible if positioned earlier due to the causal attention mask. Our ablation study in Table 5 validates this design: using directed tokens yields consistent performance gains over visual tokens across all benchmarks. In addition, to further justify this design, we conducted an additional experiment comparing placements at the beginning and end of the sequence. Results show that placing the token at the end yields superior performance, supporting our choice.
>
>
> | Place     | Science IMG | TextVQA |
> | --------- | ----------- | ------- |
> | Beginning | 70.9        | 61.4    |
> | End       | **74.3**        | **63.2**    |
>
>
> [Q2] **Visualization of multimodal misalignment**
>
>
> [A2] While visualization is restricted by the rebuttal policy, we have analyzed failure cases through systematic ablations and benchmark results. As shown in Table 1, LLaVA-v1.5 exhibits minimal performance drop when visual input is replaced with a black image, highlighting misalignment where the model overly relies on language priors. Our ablation studies in Table 6 further indicated that removing the proposed Image-to-Response Guided loss or shuffle learning leads to performance degradation across vision-intensive benchmarks (e.g., SciQA drops from 74.3\% to 72.3\%), indicating residual multimodal gaps. We will incorporate more visualization in our revised versions.
>
>
> [Q3] **The shuffle learning tasks may not fully represent semantic misalignment encountered in real-world scenarios.**
>
>
> [A3] While the tasks are constructed in a controlled manner, they are inspired by well-established techniques in vision learning, where solving shuffled inputs (e.g., jigsaw puzzles) has been widely used to improve spatial reasoning and feature generalization [1, 2, 3]. Our adaptation of this paradigm to multimodal settings aims to explicitly address real-world misalignment issues, as shown in Table 1, where LLaVA performs similarly with or without visual input. By forcing the model to reconstruct the correct order of image patches or textual segments, our method strengthens the visual-textual grounding. This is further supported by the performance improvement on real-world benchmarks (Table 3), indicating the effectiveness of shuffle learning beyond synthetic settings.
>
>
> [Q4] **The main components are individually incremental, and the novelty lies more in the combination than in fundamental algorithmic innovation.**
>
>
> [A4] We respectfully but strongly disagree with the reviewer’s assessment. The novelty of our work lies in the synergistic integration of shuffle learning, directed tokens, and guided attention into a unified framework that addresses the underexplored challenge of modality misalignment in LMMs. Our approach spans both image and text modalities through image order reconstruction and text order reconstruction tasks during pre-training. It continues with image order reconstruction during fine-tuning, enabling consistent cross-modal alignment across training stages. The proposed Image-to-Response Guided Loss further enforces visual grounding via attention supervision, a mechanism not explored in prior LMMs. As reflected in performance gains across 10+ real-world benchmarks (Tables 2–3), this integrated design delivers practical impact beyond incremental additions. Moreover, Reviewer 2 highlights the elegance of the directed token under causal attention; Reviewer 3 notes the effectiveness of coupling it with shuffle learning; and Reviewer 4 affirms that our combination of tasks and losses yields consistent improvements, even when model backbones vary.
>
>
> **References**
>
>
> [1] Carlucci, et al. Domain generalization by solving jigsaw puzzles. CVPR, 2019.
>
>
> [2] Noroozi, et al. Unsupervised learning of visual representations by solving jigsaw puzzles. ECCV. 2016.
>
>
> [3] Truong, et al. DirecFormer: A Directed Attention in Transformer Approach to Robust Action Recognition. CVPR, 2022.

---

> > ### Comment · Reviewer_oM4U · 2025-08-06
> > **Official comments by reviewer oM4U**
> >
> > Thank you to the authors for the rebuttal, which addressed my concerns. Therefore, I will keep my score unchanged.

---

> > > ### Author Response · Authors · 2025-08-06
> > >
> > > Dear Reviewer oM4U,
> > >
> > > Thank you very much for your positive feedback and rating. We are glad to hear that your concerns have been addressed.
> > >
> > > We greatly appreciate your time and constructive feedback.
> > >
> > > Best regards,
> > >
> > > Authors

---

### Note · Authors · 2025-08-12

Dear ACs and Reviewers,

We would like to thank the reviewers and area chairs for their time, constructive feedback, and active involvement in the rebuttal process. Their engagement has been invaluable in helping us clarify our work and improve the manuscript.

We are glad that our rebuttal has effectively addressed most of the reviewers’ concerns, resulting in an improvement in the overall ratings. We also greatly appreciate the thoughtful discussions with the reviewers during this stage. Although there was a reviewer who was less active in the discussion period, we greatly value their initial feedback and thoughtful consideration of our work.

Our key contributions include the introduction of directed tokens and a novel shuffle-based learning paradigm for both image and text modalities, which explicitly promotes stronger multimodal alignment. We further propose an Image-to-Response Guided Loss that enhances visual understanding by reinforcing attention from visual inputs to textual outputs. Together, our proposed approach leads to state-of-the-art performance across a broad range of multimodal benchmarks, demonstrating the effectiveness of our approach.

These contributions were acknowledged by the reviewers as significant and impactful in advancing multimodal learning. In addition, we sincerely thank the reviewers for their constructive suggestions, which we will incorporate into the final version of the paper to improve its quality further.

Best regards,

Authors

---

### Decision · Program_Chairs · 2025-09-17

**Decision:**

Accept (poster)

**Comment:**

This submission received all four positive scores from the reviewers (4xBA). The reviewers especially appreciated simplicity and clear motivation of the proposed approach, together with thorough evaluations and state-of-the art results across a broad set of metrics. The remaining questions were addressed in the rebuttal. The final recommendation is therefore to accept.